# Geometry-Aware Edge Pooling for Graph Neural Networks

**Katharina Limbeck**[*,1,2]   **Lydia Mezrag**[*,3,4]   **Guy Wolf**[†,3,4]   **Bastian Rieck**[†,1,5]

[1]Helmholtz Munich          [2]Technical University of Munich          [3]Université de Montréal
[4]Mila - Quebec AI Institute          [5]Université de Fribourg

## Abstract

Graph Neural Networks (GNNs) have shown significant success for graph-based tasks. Motivated by the prevalence of large datasets in real-world applications, pooling layers are crucial components of GNNs. By reducing the size of input graphs, pooling enables faster training and potentially better generalisation. However, existing pooling operations often optimise for the learning task at the expense of discarding fundamental graph structures, thus reducing interpretability. This leads to unreliable performance across dataset types, downstream tasks and pooling ratios. Addressing these concerns, we propose novel graph pooling layers for structure-aware pooling via edge collapses. Our methods leverage diffusion geometry and iteratively reduce a graph's size while preserving both its metric structure and its structural diversity. We guide pooling using *magnitude*, an isometry-invariant diversity measure, which permits us to control the fidelity of the pooling process. Further, we use the *spread* of a metric space as a faster and more stable alternative ensuring computational efficiency. Empirical results demonstrate that our methods (i) achieve top performance compared to alternative pooling layers across a range of diverse graph classification tasks, (ii) preserve key spectral properties of the input graphs, and (iii) retain high accuracy across varying pooling ratios.

## 1 Introduction

Graph pooling layers are important components of GNN architectures. They are implemented alongside convolutional layers to reduce the size of graph representations during training. Pooling thus enables GNNs to scale to large and complex real-world graphs while regularising the resulting representation. However, the choice of pooling method strongly influences downstream-applications and task-performance. In fact, the question of which graph properties to preserve during pooling, just as the question on the nature and quality of graph datasets in graph learning [15], remains an ongoing debate [41, 44, 55]. It is thus crucial to design expressive, efficient, and interpretable pooling layers that are capable of reliably encoding task-relevant information while reducing the size of input graphs. Most graph pooling literature takes a node-centric view [41]. However, this focus on *node-centric* rather than *edge-centric* pooling often leads to the loss of important structural information. Common pooling methods either drop nodes or optimise for a node clustering while treating graph connectivities as a secondary objective. As visualised in Figure 1 and further explored in our work, this frequently leads to counter-intuitive pooling decisions that fail to retain key geometric structures in a graph. Addressing these concerns, topological and geometric descriptors of graphs are *uniquely* poised to interoperate structural information into graph pooling.

---

[*]These authors contributed equally to this work.
[†]These authors jointly supervised this work.

39th Conference on Neural Information Processing Systems (NeurIPS 2025).

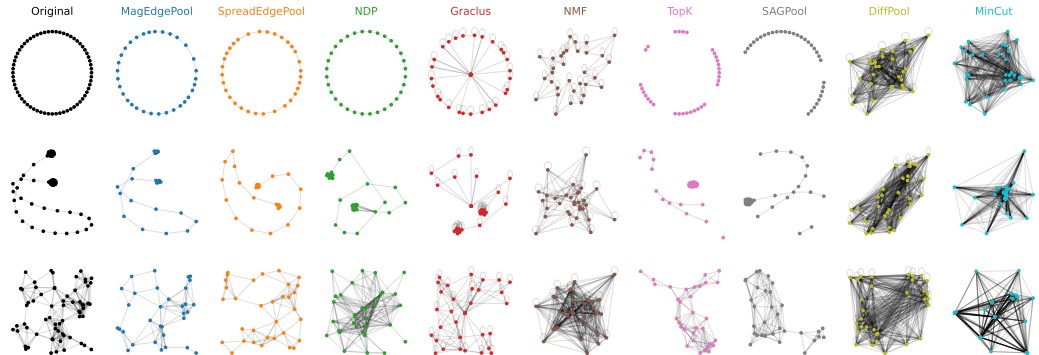

Figure 1: Examples of graphs pooled to approximately half their original size compared across pooling layers. Our proposed methods, MagEdgePool and SpreadEdgePool, respect the original graphs' geometry during pooling. Alternative approaches tend to obscure adjacency relationships to varying extents by creating counter-intuitive edges (Graclus, NMF), disconnecting entire portions of the graphs (TopK, SAGPool), or returning dense representations that do not preserve any geometric structure (DiffPool, MinCut).

Throughout this work, we treat graphs as *metric spaces* and assess their geometry via diffusion distances, which naturally work alongside message passing to effectively encode key graph structures. Motivated by ongoing research on novel geometric invariants, we find that generalised measures of size and diversity are especially promising candidates for guiding graph pooling. In particular, we use the *magnitude* of a metric space [39], which measures a graph's structural diversity, to control the loss of structural information during edge pooling. Our work is motivated by successful applications of magnitude across a range of machine learning tasks, such as the evaluation of diversity [40] for latent spaces, boundary detection for images [2], and the study of the generalisation behaviour of neural networks [3, 4]. Building up on this research, we are the first to propose the use of magnitude in the context of graph learning. We further advance on existing applications by investigating an alternative and closely-related measure, known as the *spread* of a metric space [52], to substantially improve the computational efficiency of our methods. Our main **contributions** are as follows:

- We propose MagEdgePool and SpreadEdgePool, two novel edge-contraction based pooling layers, which preserve graphs' structural diversity.
- We investigate the spread of a metric space as a faster and more stable alternative to magnitude and ensure that our algorithms can be computed efficiently.
- We evaluate our methods' capability to preserve key structural properties during pooling.
- We demonstrate that our pooling methods *consistently* perform well in graph classification tasks, achieving top accuracies among other pooling layers across a wide range of experimental setups.

## 2 Background

We briefly provide background information on graph pooling, the magnitude of metric spaces, and diffusion geometry, taking care to point out related work and how it differs from ours.

### 2.1 Graph Pooling for Graph Neural Networks

As an ongoing field of interest for graph learning, a wide range of pooling methods has been proposed, which can be divided into *global* and *hierarchical* approaches. Global and hierarchical pooling methods fulfil fundamentally different but complementary roles as distinct components of GNN architectures [32]. Global or flat pooling methods generate graph-level representations by reducing each graph to a single node and are commonly used as readout operations. Examples include mean-pool or sum-pool, which average or sum all node features, SOPool [51], which uses second-order feature information, or DKEPool [13], which learns node distributions. However, global pooling methods discard all topological information, which can reduce the expressivity of these models and decrease their performance [8, 32, 34]. By contrast, hierarchical pooling methods sequentially coarsen graph representations while reducing their size, and are frequently used in alternation with

intermediate convolutional or message passing layers [41]. By reducing both features and adjacencies, hierarchical methods have the ability to preserve graphs' inherent geometry allowing GNNs to learn across multiple coarsening levels.

Research has traditionally focused on *node-based* hierarchical pooling via node clustering or node dropping. Amongst node drop pooling methods, Node Decimal Pooling (NDP) [10] is non-trainable, while TopK [12, 28] and SAGPool [36], are trainable approaches. However, all these methods inevitably lose information because they do not select all vertices but remove entire sections of the graph during pooling [41]. Node clustering approaches are similarly either non-trainable, such as Graclus [19] and Non-Negative Matrix Factorization (NMF) [6], or trainable, such as MinCUT [9] and DiffPool [56], which output dense graph representations. In comparison to node drop pooling, clustering-based methods usually require high memory costs [41]. Overall, trainable methods have the potential to better optimise for a specific objective, but might overfit to the task at hand, especially for smaller datasets. By contrast, non-trainable methods can act as a stronger inductive bias on the underlying graph representation, and do not introduce additional trainable parameters or optimisation objectives into the GNN [32]. *Edge-based* pooling methods have been studied less extensively than node-centric pooling [20, 35] despite their promising potential for encoding connectivities in a more faithful and natural manner. EdgePool [20], the most successful edge-based method, uses iterative edge-contraction based on edge scores, which are learnt from features of adjacent nodes. However, EdgePool always pools graphs to half their size, making it less flexible than more adaptive methods. Moreover, learning the edge scores during training is computationally expensive [8] necessitating the development of faster and more efficient edge pooling approaches. Further, EdgePool does not explicitly consider a graph's topology, beyond posing a constraint on not contracting adjacent edges. Thus, selecting edges based on node features can lead to counter-intuitive decisions during pooling, for example by retaining local structures in strongly-connected communities even at high pooling ratios [20]. Interpretability remains a leading concern [41], and non-trainable approaches are capable of being well-performing and traceable baseline pooling methods [32]. We therefore find that there is strong potential for designing geometry-aware edge pooling operations that make interpretable decisions on which aspects of the graph to retain. Figure 1 further illustrates how many of the aforementioned pooling methods inherently fail to preserve key graph structures even for simple toy examples. Trainable methods in this overview were optimised for a spectral loss following Grattarola et al. [32], but this does *not* ensure interpretable preservation of graphs' underlying geometry. Addressing these shortcomings, it is of interest to leverage alternative tools from computational geometry, which allow us to quantify the qualitative difference between graphs. While pooling based on spectral properties has been investigated extensively [10, 32], alternative geometric invariants like curvature or persistent homology have only been explored more recently [25, 55], and have not yet been applied to edge pooling specifically.

## 2.2 The Magnitude and Spread of Metric Spaces

Magnitude is an invariant of (finite) metric spaces that measures the 'effective size' of a space. It is a measure of entropy and diversity [39] that has first been proposed in theoretical ecology [48]. Since its mathematical formalisation by Leinster [37], magnitude has been connected to numerous key geometric invariants, such as entropy, curvature, density, volume, and intrinsic dimensionality [39]. Because of its intriguing theoretical properties magnitude has received increasing interest for machine learning tasks [40]. However, the magnitude of graphs [38], despite being a strong graph invariant, has not yet found its way into applications.

Throughout this paper, we consider an undirected finite graph $G = (X, E)$ with $n$ nodes as a *finite metric space*, consisting of the node set $X$ equipped with a *metric* $d \colon X \times X \to \mathbb{R}_{\geq 0}$. Its similarity matrix $\zeta_X \subseteq \mathbb{R}^{n \times n}$ is defined by $\zeta_X(x, y) = e^{-d(x,y)}$ for $x, y \in X$. This allows us to introduce similarity-dependent notions of the diversity of metric spaces. To this end, we define a *weighting* on the metric space $(X, d)$, which is a vector $w \in \mathbb{R}^n$ such that $\zeta_X w = \mathbb{1}$, where $\mathbb{1}$ is the column vector of ones. Whenever such a weighting exists, the *magnitude* of the metric space $(X, d)$ is *uniquely* defined by $\mathrm{Mag}(X) = \sum_{i=1}^n w(i)$. This is guaranteed if $\zeta_X$ is positive definite, which essentially means that $\zeta_X$ is invertible. A finite metric space with positive definite similarity matrix is called *positive definite* [43]. Metric spaces of negative type are positive definite [39]; this includes $\mathbb{R}^n$ equipped with the Euclidean distance [37], effective resistance distances [18], and diffusion distances [14]. Subsequently, we will refer to the magnitude of a graph $G$ as the magnitude of its

associated metric space $(X, d)$. That is, we define the *magnitude of a graph* as

$$\text{Mag(G)} = \sum_{x,y \in X} \zeta_X^{-1}(x, y). \tag{1}$$

Closely related to magnitude, the *spread* of a metric space is another measure of 'size' introduced by Willerton [52]. Given a metric graph $G$ with the graph metric $d$, its *spread* is defined by

$$\text{Sp}(G) := \sum_{x \in X} \frac{1}{\sum_{y \in X} e^{-d(x,y)}}. \tag{2}$$

As diversity measures on graphs, both *magnitude* and *spread* summarise the number of distinct sub-communities in a network based on the distance metric and degree of similarity between nodes. This view on structural diversity naturally aligns with our goal of contracting redundant graph structures during pooling. Throughout this work, we investigate to what extent spread is a valid alternative to magnitude. This is motivated by the fact that computing magnitude in practice either requires inverting a matrix, solving a system of linear equations [40], or resort to approximations [5], which can be computationally expensive and numerically unstable. Metric-space *spread* in comparison can be computed given *any* distance (obviating the requirement of metric spaces of negative type), making it much more versatile [52]. Moreover, as the sum of reciprocal mean similarities, spread can be calculated or approximated [23] much more efficiently than magnitude and does not require inverting a matrix. Although spread has been studied less extensively, there are strong reasons to assume that it shares the same advantages as magnitude. In fact, for a positive definite metric space $X$, we have $\text{Sp}(X) \leq \text{Mag}(X)$ [52, Theorem 2]. Moreover, magnitude and spread coincide for finite homogeneous metric spaces [52, Theorem 3], such as the ring graph in Figure 1. In practice, as we further explore in Appendix D.1, the magnitude and spread of graphs from real-world datasets, such as NCI1, exhibit nigh-perfect correlation when computed based on diffusion distances, underlining the strong connection between the two quantities.

## 2.3 Diffusion Geometry on Graphs

In this work, we use diffusion distances to compute magnitude and spread on graphs. This is motivated by their desirable theoretical properties and the capability of diffusion to aid and act along message passing in GNNs: Diffusion operators are closely associated to random walks and are efficient at identifying important structures in complex geometries while preserving local and non-linear structures [30]. The key idea is that the eigenvectors of the Markov matrices can be thought of as coordinates for the underlying graph structure [14]. This provides a vector-space representation of the graph that can be used to assess the dissimilarity between nodes.

We now briefly detail the type of diffusion distance used throughout this paper. Consider a graph $G$ and its adjacency matrix $A$. Let $D$ be the diagonal degree matrix whose diagonal entries $D_{ii} = \sum_{j=1}^{n} A_{ij}$ equal the degree of each vertex. The symmetrically *normalised adjacency matrix* $\hat{A} := D^{-\frac{1}{2}} A D^{\frac{1}{2}}$ is a *Markov transition matrix* and represents the probability of moving from one vertex to another. The *normalised graph Laplacian* is defined as $\hat{L} = I - \hat{A} = D^{-\frac{1}{2}}(D - A)D^{-\frac{1}{2}}$. Since $\hat{L}$ is symmetric and positive definite, it has positive eigenvalues $2 \geq \lambda_0 > \lambda_1 > \lambda_2 > \cdots > \lambda_{n-1} \geq 0$ with eigenvectors $\{\psi_l\}_l$. This provides a natural embedding of the graph $G$ in Euclidean space given by:

$$\Phi(x) = (\lambda_1 \psi_1(x), \cdots, \lambda_{n-1} \psi_{n-1}(x)) \text{ for } x \in X. \tag{3}$$

The *diffusion distance* is then defined by the $l^2$-norm, i.e.,

$$d(x, y) = \|\Phi(x) - \Phi(y)\|_2 \text{ for } x, y \in X. \tag{4}$$

**Theorem 1.** *Any finite metric space $(X, d)$ endowed with the diffusion distance is positive definite.*

As a consequence of Theorem 1, the magnitude of any metric graph equipped with this diffusion distance is well defined. Leveraging diffusion distances for our methods has further benefits. The normalised graph Laplacian, which works with the relative connectivity between nodes, is robust to varying node degrees and graph sizes. This ensures that diffusion distances are on comparable scales across graphs, which enables us to compute and compare magnitude and spread directly. We note, however, that our pooling methods are flexible and can be applied to a wider range of alternative distances or similarities between nodes, which can be tailored to an application domain.

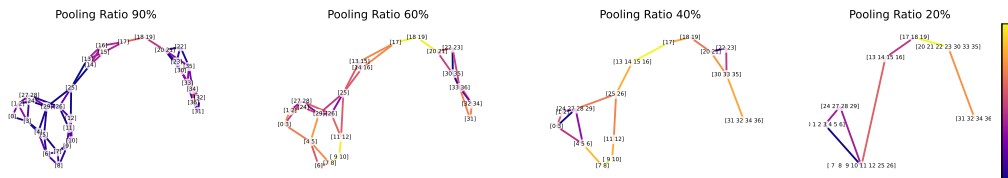

Figure 2: Illustrating our proposed pooling method, MagEdgePool, on a graph from the ENZYMES dataset across varying pooling ratios. Each edge is coloured by its magnitude difference, which measures the impact its contraction would have on the graph's structural diversity. Edges with low magnitude differences are most redundant for the graph's geometry and are collapsed first.

## 3 Methods

We first describe our magnitude-guided graph pooling methods in Section 3.1 while providing an in-depth theoretical analysis in Section 3.2.

### 3.1 Magnitude-Guided Graph Pooling

At the heart of our approach, we use magnitude or spread to monitor and control structural changes in the graph during edge contraction pooling. Edge contraction is chosen as a pooling operation because it respects graph connectivity, while outputting a sparsely-connected graph representation. Conceptually, edge pooling thus aligns well with the goal of making *minimal* changes to the graph and keeping its diversity and geometry as unchanged as possible. Intuitively, pooled graphs with comparable magnitude will be similar in terms of effective size. That is, the effective number of distinct communities in the graphs are deemed similar based on their diffusion distance, i.e. based on the information flow between vertices.

Formally, let $G = (X, E)$ be a graph and denote by $G/e$ the graph resulting from the contraction of the edge $e \in E$ in $G$. We choose a pooling ratio $r \in (0, 1]$ and aim to reduce the graph to the corresponding number of nodes i.e. $k = \lfloor r \cdot |X| \rceil$ where $\lfloor \rceil$ denotes rounding to the nearest integer. Initially, we set the pooled graph $G' = (X', E') := G$. To assess which edges to contract first, we determine their importance for the graph's global geometry by computing a *selection score* for each edge, which we define as

$$s(e) = |\text{Mag}(G) - \text{Mag}(G/e)|. \tag{5}$$

That is, we calculate the difference in magnitude between the original graph and the graph for which the edge has been collapsed. In this manner, we score an edge's relevance for the graph structure by the impact its collapse would have on the graph's magnitude. In each iteration $i$, we then select the edge with the lowest magnitude difference

$$e_i \in \arg \min_{e \in E' \setminus E_c} s(e) \tag{6}$$

and assign $G' = G'/e_i$ where $E_c \subseteq E'$ is the set of all edges that are adjacent to an already contracted edge. The edge $e_i$ that we contract is selected *at random* whenever there is more than one valid option. If all edges that meet this requirement have been collapsed, but the pooling ratio is not reached, we re-compute the edge scores on the new graph and repeat this procedure. We stop if the pooling ratio is reached, i.e. $|X'| = \lfloor r \cdot |X| \rceil$, or if there are no edges left in the reduced graph, i.e. $E' = \emptyset$. This approach allows us to flexibly reduce graphs to any desired size.

Edge pooling then gives a hard assignment of nodes, where each vertex is assigned to a single super-node in the output graph based on the neighbours it has been merged with. To compress the node features, we *average* the features of any node that contributed to a pooled super-node. This ensures that information on all nodes' features is preserved during pooling. Note that the feature aggregation function could easily be modified to use sum pooling instead of mean pooling, for example. Restricting the number of times a vertex can be merged is enforced to prevent our methods from collapsing entire portions of a graph early in the pooling process. This enables a more uniform pooling across the graph, which aids feature preservation and expressivity.

**In a nutshell:** Our methods assume that the *most redundant edges* will be those whose removal would change the diversity of the graph the least. Magnitude informs us about the global importance of an edge. This implies that we want to start by merging edges from well-connected communities whenever their contraction does not notably change the graph's effective size.

As illustrated in Figure 2, our pooling method ensures that globally influential edges will be collapsed *late* in the pooling progress. For example, the edge which bridges the two parts of this enzyme graph is scored more highly and thus merged later than well-connected cliques. Redundant local structures are collapsed first, and the overall (diffusion) geometry is respected. For the graph in Figure 2, this ensures that characteristic features of the enzyme, such as the *cycle*, are preserved across the pooling process. We provide a pseudocode implementation of our algorithm in Appendix C.4. Notice that we may use, *mutatis mutandis*, spread in lieu of magnitude. Whenever we use magnitude to compute the edge scores, we denote our algorithm by MagEdgePool, else we use the moniker SpreadEdgePool.

## 3.2 Theoretical Analysis

We now present theoretical properties of our pooling methods. First, we highlight fundamental invariances and properties of magnitude and spread used to design our algorithms. Further, we provide a bound on the difference in magnitude during pooling by the difference in spread, demonstrating the close relationship between the two. For a complete list or theorems and proofs, please refer to Appendix B.3.

**Additivity for disjoint graphs.** An important property of magnitude is that it behaves like the cardinality of sets. This behaviour is useful when computing the magnitude of a disconnected graph by splitting the problem into calculating the magnitude of the disconnected components.

**Theorem 2.** *Consider a graph $G = G_1 \sqcup G_2$ consisting of the disjoint union of two graphs $G_1$ and $G_2$. Then $Mag(G) = Mag(G_1) + Mag(G_2)$.*

**Isomorphism invariance.** Our pooling layers are invariant under isometries of the input graph provided that the edge choice at each iteration is deterministic whenever the edge scores coincide. This is a consequence of the following result.

**Theorem 3.** *For* isomorphic *graphs $G_1, G_2$, we have $Mag(G_1) = Mag(G_2)$ and $Sp(G_1) = Sp(G_2)$.*

**Edge contraction on graphs.** Edge contraction is the main operation of our pooling methods and can be considered as a map $f$ between graphs. The following result provides a sufficient condition to ensure that $f$ remains compatible with the metric structure.

**Theorem 4.** *Consider an edge-contraction map $f : (G_1, d_1) \rightarrow (G_2, d_2)$ between positive definite metric graphs. If the map is 1-Lipschitz, i.e. $d_2(f(v_1), f(v_2)) \leq d_1(v_1, v_2) \ \forall v_1, v_2 \in X_1$, then $Mag(G_2) \leq Mag(G_1)$.*

Starting from an initial metric graph $G = (X, E)$, a 1-Lipschitz edge-contraction map $f$ yields a sequence of graphs $\{G_i\}_{i=1}^k$, where $k$ is the number of edges that have been contracted. This sequence can be constructed as described by our algorithms in Section 3.1. We refer to the graphs resulting from the $k^{th}$ edge-contraction using MagEdgePool as $G^{(k)}$. Let $\Delta^{(k)}\mathrm{Mag}(G) = |\mathrm{Mag}(G^{(k-1)}) - \mathrm{Mag}(G^{(k)})|$ and let $\Delta^{(k)}\mathrm{Sp}(G) = |\mathrm{Sp}(G^{(k-1)}) - \mathrm{Sp}(G^{(k)})|$.

**Bounding magnitude by spread.** We track the difference in magnitude and spread throughout the edge contraction process detailed above. This allows us to propose an inequality that describes the relation between the difference of magnitude and spread during pooling. The bound then demonstrate the close conceptual relationship between SpreadEdgePool and MagEdgePool.

**Theorem 5.** *Consider a positive definite metric graph $G$ with positive weights. Assume that the edge-contraction maps describing MagEdgePool and SpreadEdgePool induce distance decreasing surjections on the vertex sets. If $|Mag(G^{(k-1)}) - Sp(G^{(k)})| \leq C\Delta^{(k)}Sp(G)$, then*

$$\Delta^{(k)}Mag(G) \leq 3C\Delta^{(k)}Sp(G).$$

**Expressivity.** Studying the expressive power of GNNs and their ability for distinguishing non-isomorphic graphs offers theoretical insights for understanding the theoretical capabilities of pooling operators. Bianchi and Lachi [8] state sufficient condition for a pooling layer to preserve the expressive power of the preceding message-passing (MP) layers. As demonstrated in Appendix B.4, MagEdgePool and SpreadEdgePool satisfy these conditions ensuring expressivity.

**Computational Complexity.** The time complexity of our pooling methods is independent of the GNN and is dominated by the cost of computing the edge scores in Equation (5). Given a graph $G = (X, E)$, magnitude has time complexity $O(|X|^3)$. In comparison, spread has time complexity $O(|X|^2)$ and can be more efficiently approximated [23]. Spread thus offers a considerably faster alternative. Now, let $O(C_d)$ be the time complexity of computing the metric on $G$. The time complexity of SpreadEdgePool is dominated by $O(|E|(C_d + |X|^2 + \log|E|))$ and MagEdgePool by $O(|E|(C_d + |X|^3 + \log|E|))$. We note that on large graphs, it is possible to speed up the distance computations further to ensure scalability. See Appendix B.2 for a full description of computational costs and Appendix D.2 for an empirical evaluation, which shows that our algorithm performs on-par with existing pooling methods across the datasets evaluated throughout this work.

## 4 Experimental Results

Across our experiments, we address four key tasks, namely (i) graph classification performance, (ii) graph structure preservation during pooling, (iii) performance across varying pooling ratios, and (iv) performance at graph property regression.

### 4.1 Graph Classification

The primary aim of using graph pooling layers is to preserve task-relevant information while reducing computational costs. In particular, useful pooling layers guarantee good performance across a wide range of different datasets, thus capturing essential information for the task at hand. We thus investigate how well our aim to preserve structural diversity during edge pooling translates to practical performance at graph classification tasks. Note that our goal is not to reach state-of-the-art accuracies on all tasks, but to benchmark the performance gain or loss of different pooling operators.

**Experimental Setup.** We evaluate 8 different graph datasets, as detailed in Appendix C.2. Whenever node features are not available, we use node degree as an input feature. To ensure a fair comparison with alternative pooling methods, we follow the experimental setup by Grattarola et al. [32] and guidance by Errica et al. [24] for fair model comparison. Specifically, we plug in each pooling layer into the model architecture specified by Grattarola et al. [32], which is of the following form:

$$\text{MLP}(\mathbf{X}) \to \text{GNN}(\mathbf{X}, \mathbf{A}) \to \text{POOL}(\mathbf{X}, \mathbf{A}) \to \text{GNN}(\mathbf{X}, \mathbf{A}) \to \text{GlobalSum}(\mathbf{X}) \to \text{MLP}(\mathbf{X})$$

The model includes pre-processing and post-processing MLPs with 2 layers, 256 hidden units, ReLU activation, and batch normalization. $\text{GNN}(\mathbf{X}, \mathbf{A})$ refers to a graph neural network layer, more specifically a general convolutional layer [57] with parameters chosen according to the best results achieved by You et al. [57]. As an intermediate layer, $\text{POOL}(\mathbf{X}, \mathbf{A})$ corresponds to a specific pooling layer. All pooling layers are configured to pool each graph to around 50% of nodes. We also compare with 'No Pooling,' the same model architecture without any pooling layers. We use 10-fold stratified cross-validation and further partition the training data into 90% training and 10% validation data while keeping the labels balanced between splits. Finally, we report the best test accuracy of each model trained using Adam with a cross-entropy loss (batch size 32, learning rate 0.0005, and early stopping based on the validation loss with a patience of 50 epochs). Further details are described in Appendix C.5.3.

**Classification Results.** Table 1 reports the mean and standard deviation of the test accuracy achieved by different pooling methods. We furthermore highlight which methods do *not* perform statistically significantly different from the best model (using pairwise Wilcoxon signed-rank tests applied to the accuracy scores and employing Holm–Bonferroni correction at a significance threshold of $p = 0.05$), thus permitting us to identify pooling methods that achieve top performance. Notably, both MagEdgePool and SpreadEdgePool achieve the *best mean ranks* across datasets in terms of their

Table 1: Mean and standard deviation of the graph classification accuracy. The best-performing model is marked in bold. All models that did not perform significantly different from the best model are coloured green. The rightmost column shows the mean rank of each pooling method across datasets.

| Method | ENZYMES | PROTEINS | Mutagenicity | DHFR | IMDB-B | IMDB-M | NCI1 | NCI109 | Mean Rank |
|---|---|---|---|---|---|---|---|---|---|
| No Pooling | 87.3 ± 2.5 | 73.8 ± 0.8 | 80.1 ± 1.3 | 71.4 ± 1.9 | 69.7 ± 0.7 | 46.0 ± 0.7 | 76.5 ± 1.8 | 74.3 ± 2.0 | - |
| MagEdge | 91.5 ± 3.2 | 76.4 ± 3.9 | **77.5 ± 2.7** | 88.0 ± 3.8 | 72.4 ± 1.7 | 47.4 ± 1.7 | 72.7 ± 2.4 | 73.0 ± 3.3 | **2.4** |
| SpreadEdge | **92.8 ± 1.6** | 75.1 ± 3.1 | 76.0 ± 4.0 | **90.7 ± 3.8** | 71.8 ± 1.5 | 47.3 ± 1.7 | 73.4 ± 2.5 | 71.8 ± 1.8 | **3.0** |
| NDP | 92.2 ± 1.6 | 73.7 ± 3.9 | 73.4 ± 3.1 | 79.6 ± 4.4 | **73.3 ± 2.0** | 47.3 ± 2.5 | 70.6 ± 2.2 | 70.0 ± 2.2 | 3.6 |
| Graclus | 91.3 ± 3.7 | **76.6 ± 3.7** | 72.5 ± 2.0 | 64.4 ± 5.8 | 71.9 ± 1.5 | **49.3 ± 2.4** | 68.8 ± 1.4 | 69.5 ± 2.1 | 5.6 |
| NMF | 78.6 ± 8.0 | 73.0 ± 8.1 | 71.0 ± 4.7 | 66.5 ± 7.7 | 69.4 ± 2.5 | 43.3 ± 1.7 | 71.0 ± 3.7 | 72.0 ± 5.3 | 6.0 |
| TopK | 82.2 ± 7.5 | 73.2 ± 1.4 | 75.8 ± 4.7 | 68.9 ± 3.0 | 68.9 ± 1.5 | 45.6 ± 1.0 | **75.3 ± 2.4** | 73.9 ± 3.3 | 4.9 |
| SAGPool | 82.4 ± 4.5 | 73.8 ± 1.3 | 76.0 ± 2.6 | 69.9 ± 3.0 | 69.1 ± 0.6 | 45.7 ± 0.5 | 74.3 ± 2.8 | **74.0 ± 2.3** | 4.7 |
| DiffPool | 74.0 ± 5.7 | 68.9 ± 2.0 | 68.4 ± 1.9 | 79.8 ± 3.2 | 68.3 ± 0.8 | 44.4 ± 0.8 | 68.9 ± 1.0 | 68.3 ± 1.9 | 7.6 |
| MinCut | 80.2 ± 6.6 | 75.6 ± 1.3 | 70.9 ± 1.5 | 63.8 ± 3.7 | 69.3 ± 0.7 | 46.1 ± 0.8 | 66.7 ± 1.4 | 66.9 ± 2.0 | 7.4 |

accuracy. Further, they are always among the top-performing methods across all evaluated datasets. Altogether, both their ranking and their individual accuracy scores thus demonstrate superior and consistently high performance across graph classification tasks. The performance benefits of our proposed pooling methods are most pronounced on DHFR [49], where our methods surpass even the GNN without pooling layer by around 17 percentage points. This provides evidence that the regularising effects of our pooling approach can help reduce overfitting, especially for small datasets and geometrically-rich graphs. For other biological datasets (Mutagenicity, NCI1 and NC109), our methods show competitive performance with trainable layers, indicating that the introduction of additional trainable components into the pooling layer is *not* necessary to guarantee high task performance. On ENZYMES, DHFR, IMDB-BINARY, and IMDB-MULTI, non-trainable methods generally outperform trainable pooling layers, with MagEdgePool and SpreadEdgePool consistently reaching high accuracy.

Comparing pooling methods to using no pooling, we observe that MagEdgePool and SpreadEdgePool reach similar or even higher performance across datasets. For six datasets, our diversity-guided pooling methods improve mean accuracy, indicating that pooling retains task-relevant information while aiding the generalisation capabilities of the GNN. MagEdgePool and SpreadEdgePool act as interpretable and expressive pooling transformations (see Appendix D.3), which reduce the computational costs making GNNs learn from graphs' coarsened geometry. As reported in Table 1, MagEdgePool and SpreadEdgePool achieve very similar accuracies across datasets reaching top accuracies compared to alternative pooling layers. In practice, especially for large graphs, we recommend using SpreadEdgePool due to its high predictive performance and superior computational efficiency.

## 4.2 Magnitude and Graph Structure Preservation

Motivated by the visual comparison of graphs pooled using different pooling layers from Figure 1, we next set out to investigate the link between structure preservation and task performance. Specifically, we choose NCI1, a dataset of 4,110 graphs corresponding to chemical compounds, because it has been shown to possess both informative features and task-relevant graph structures [15]. We follow the same classification procedure as before and extract the pooled graph representations after training the GNNs described in Section 4.1. Three pooling layers, MinCut, DiffPool and NMF, were removed from further comparison because they showed notably worse qualitative results for the motivating examples in Figure 1 and classification performance in Table 1. NDP and Graclus are evaluated across fewer pooling ratios than more adaptive methods, because they pool graphs to around half their size at every step. To assess graph structure preservation, we use the spectral distance defined as $\sqrt{\sum_{k=1}^{K}(\lambda_k - \lambda'_k)^2}$, i.e. the $l_2$-norm between the eigenspectra of the normalised Laplacians of the original and the pooled graphs [53]. We also report the magnitude difference between graphs to evaluate the preservation of structural diversity.

MagEdgePool and SpreadEdgePool exhibit small spectral distances across pooling ratios as visualised in Figure 3. This indicates that contracting edges guided by structural diversity *preserves* key spectral properties. Our methods also demonstrate low magnitude differences confirming that they perform as intended. In fact, the structure preservation scores for MagEdgePool and SpreadEdgePool coincide

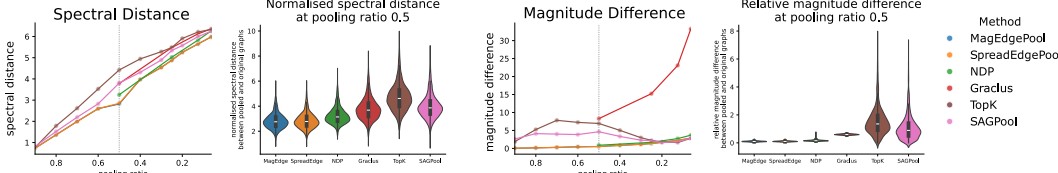

Figure 3: Structure preservation for all graphs in the NCI1 dataset across varying pooling ratios. Left: The spectral distance between the normalised Laplacians of the original and the pooled graphs. Right: The relative difference in magnitude, which summarises the proportional difference in structural diversity after pooling. Violin plots show the variability across graphs at pooling ratio 0.5.

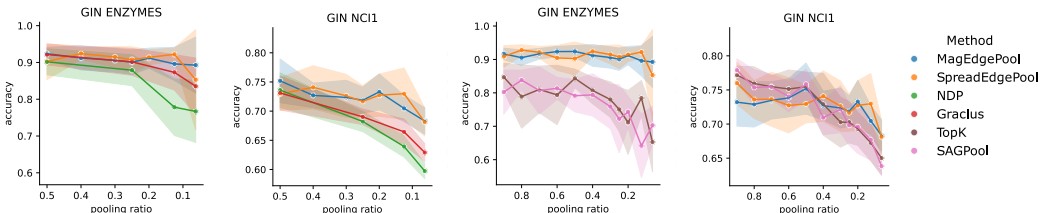

Figure 4: Classification performance across varying the pooling ratio for different pooling layers. Pooling is applied as part of a GIN architecture. Results are shown for the ENZYME and NCI1 datasets. Lines show the mean and shaded areas the standard deviation of the test accuracy.

almost perfectly, giving empirical evidence that spread offers an alternative to magnitude. Figure 3 indicates that preserving magnitude corresponds to lower spectral distances and better retention of spectral properties. This link supports our motivation of guiding pooling by magnitude.

Alternative pooling layers fail to effectively preserve graphs' structural properties during both qualitative and quantitative comparisons to varying extents. Node decimal pooling (NDP) [10] was specifically designed to preserve spectral properties during pooling. However, it still reaches both higher spectral distances and higher magnitude differences than MagEdgePool on average across pooling ratios, as visualised in Figure 3. Finally, the sparse pooling layers Graclus, TopKPool, and SAGPool, all show higher spectral distances than our approach. This difference is even more pronounced in terms of magnitude differences, where all these three methods demonstrate high distortion of the underlying metric space diversity. These findings are repeated for further datasets (see Figure S.7 in the appendix) and agree with the qualitative comparisons between graphs pooled using different pooling layers as visualised in Figure 1. We thus conclude that MagEdgePool and SpreadEdgePool successfully encode graphs' coarsened geometry during pooling, surpassing alternative pooling methods.

## 4.3 Pooling Ratio and Task Performance

From Table 1, we observe that it is possible to reach very high performance on benchmark datasets even while pooling each graph to half its size. Based on this, we further investigate how the pooling ratio influences pooling layers and their classification performance. We consider two datasets, NCI1 and ENZYMES, which contains 600 graphs representing protein tertiary structures from 6 classes of enzymes and is selected as an example of a multi-class prediction task. We keep the same experimental setup described in Section 4.1, but use GIN layers instead of general convolutional layers to further assess whether the trends in performance differ across model architectures. Figure 4 reports the accuracy achieved by each pooling layer for varying pooling ratios. Notably, we observe that MagEdgePool and SpreadEdgePool consistently reach very high test accuracies even at low pooling rations. Meanwhile, the performance of other non-trainable methods drops notably more when graphs are pooled to up to 6.25% of their original size showing that they fail to preserve task-relevant information for both ENZYMES and NCI1. We note that the trainable pooling layers, TopK and SAGPool, reach higher or comparable performance on NCI1 for pooling ratios above 50%, but decrease in accuracy for more extreme pooling ratios. They consistently perform worse on ENZYMES, indicating that they distort important graph features or key graph structures during

pooling. MagEdgePool and SpreadEdgePool in comparison reach top performance and lower decreases in accuracy across varying pooling ratios demonstrating their potential to offer reliable, interpretable and stable pooling operations. Overall, the reported accuracies in Figure 4 agree with results in Table 1 indicating that our observations hold for varying choices of GNN layers. We thus find across experiments that MagEdgePool and SpreadEdgePool constitute useful general-purpose pooling approaches, demonstrating their capability to faithfully encoding graphs' geometry, which ensures stable performance across pooling ratios, datasets and GNN architectures.

## 4.4   Graph Regression

We further apply our pooling methods to graph regression tasks. We modify the GNN architecture as described in Appendix C.5.4 and plug in varying pooling layers. Table 2 reports the RMSE on the test dataset across pre-defined data splits and ten random seeds for three datasets from OGB [33]. Overall, these results confirm that SpreadEdge-Pool and MagEdgePool constitute useful pooling operations and reach comparatively low RMSEs across these three regression tasks. Our pooling methods successfully retain or improve task performance, indicating their ability to preserve task-relevant information.

Table 2: RMSE on the test data for different pooling methods.

| Method | MolEsol | MolFreeSolv | MolLipo |
|---|---|---|---|
| No Pooling | $1.44 \pm 0.10$ | $2.98 \pm 0.78$ | $0.98 \pm 0.30$ |
| MagEdge | $1.47 \pm 0.11$ | $2.81 \pm 0.30$ | $0.91 \pm 0.23$ |
| SpreadEdge | $1.58 \pm 0.15$ | $2.83 \pm 0.24$ | $0.91 \pm 0.20$ |
| NDP | $1.54 \pm 0.18$ | $3.17 \pm 0.29$ | $0.82 \pm 0.09$ |
| Graclus | $1.47 \pm 0.13$ | $2.99 \pm 0.27$ | $0.85 \pm 0.17$ |
| NMF | $2.37 \pm 0.77$ | $15.06 \pm 8.83$ | $0.85 \pm 0.04$ |
| TopK | $1.68 \pm 0.10$ | $3.25 \pm 0.54$ | $1.07 \pm 0.25$ |
| SAGPool | $1.66 \pm 0.18$ | $2.77 \pm 0.18$ | $1.03 \pm 0.20$ |
| DiffPool | $1.71 \pm 0.14$ | $5.78 \pm 4.55$ | $2.35 \pm 2.57$ |
| MinCut | $1.88 \pm 0.51$ | $4.28 \pm 0.32$ | $1.22 \pm 0.19$ |

## 4.5   Computational Efficiency

Finally, we evaluate the computational efficiency and scalability of pooling operators (see Appendix D.2 for details). Our approaches scale reasonably well to the datasets evaluated in this study, both in terms of runtime and memory requirements, thus remaining applicable for graphs with up to few hundreds of nodes. Experiments confirm that SpreadEdgePool is notably faster than MagEdge-Pool and that distance approximations can be used to speed up the edge score computations used for pooling. Further, we find that pre-computing allows for notably lower GNN training costs compared to trainable methods; when comparing our methods to EdgePool, our methods exhibit superior computational efficiency during training while ensuring similarly high classification performance.

## 5   Discussion

Despite its advantageous properties, our method exhibits certain *limitations*: We implicitly require redundancy and homophily in the graph representation to make it amenable to geometry-guided pooling. Our methods further assume that preserving graph structure is *beneficial* to the learning task. Otherwise, edge pooling still aggregates features faithfully (as evaluated in Appendix D.3), but the geometric objective might not be necessary to ensure high performance (Appendix D.7 provides an extended discussion and ablation study on the importance of preserving graph structure or preserving expressivity during pooling). Moreover, our algorithm relies on efficient distance computations, and we only explore one specific case of diffusion distances, which could be sped up (see Appendix D.2) or generalised further in future work. Although the non-trainable nature of our pooling methods can be limiting, our experiments nevertheless demonstrate that trainable pooling layers do *not* guarantee higher performance on standard benchmark datasets for graph classification or regression.

Across experiments, we thus find that MagEdgePool and SpreadEdgePool constitute useful general-purpose pooling approaches: They are competitive compared to state-of-the-art pooling layers for graph classification or regression tasks, and perform well for a wide range of datasets and pooling ratios. Guiding edge pooling to preserve graphs' structural diversity successfully encodes key graph properties, ensuring stable and interpretable performance. Further, we overcome a major limitation of computing magnitude on large graphs by proposing spread as a substantially faster and closely-related alternative, which has the potential to aid research in geometric deep learning and efficient diversity evaluation beyond the scope of this paper. For future work, we plan on applying our methods with alternative graph distances, to scale computations to large-scale graphs, or to determine an ideal pooling ratio automatically. Our methods are available as a Python package on GitHub.[1]

---

[1] `https://github.com/aidos-lab/mag_edge_pool` available under a BSD 3-Clause License.

## Acknowledgements

The authors are grateful for the stimulating discussions with the anonymous reviewers and the area chair, who believed in the merits of this work. K.L. is supported by the Helmholtz Association under the joint research school 'Munich School for Data Science (MUDS).'

## Funding Disclosure

This work was partially funded by the Helmholtz International Lab Causal Cell Dynamics InterLabs-0029 grant by the Initiative and Networking Fund of the Hermann von Helmholtz-Association Deutscher Forschungszentren e.V. [K.L.], Mila EDI scholarships [L.M.], Humboldt Research Fellowship, CIFAR AI Chair, NSERC Discovery grant 03267, FRQNT grant 343567, and NSF grant DMS-2327211 [G.W.]. This work has received funding from the Swiss State Secretariat for Education, Research, and Innovation (SERI). The content provided here is solely the responsibility of the authors and does not necessarily represent the views of the funding agencies.

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

# Appendix (Supplementary Materials)

# A    Technical Appendices and Supplementary Material

To elaborate on the results reported in our main paper, we first detail extended theoretical results and proofs for our theoretical contributions. In particular, we investigate computational complexity as well as the relationship between magnitude and spread. Next, we detail our experimental evaluation, the assets used for our experiments, and the algorithm describing our pooling methods. Finally, we report extended results on the experiments included in our main paper.

# B    Theoretical Analysis

This section details full proofs and extended explanations for the mathematical theory introduced in Section 2 and the theoretical analysis of our pooling methods described in Section 3.2.

## B.1    Diffusion Distances

As detailed in Section 2.3, the *diffusion distance* is defined by

$$d(x, y) = \|\Phi(x) - \Phi(y)\|_2 \text{ for } x, y \in X. \tag{7}$$

**Theorem 1.** *Any finite metric space* $(X, d)$ *endowed with the diffusion distance is positive definite.*

*Proof.* By definition of the diffusion distance, the map $\Phi : X \to \mathbb{R}^{N-1}$ in Equation (3) defines an isometry $(X, d) \hookrightarrow l_2^{N-1} := (\mathbb{R}^{N-1}, d_2)$, where $d_2$ is the metric induced by the $l_2$-norm. Finally, by Theorem 2.5.3 in Leinster [37], subsets of Euclidean space $l_2^{N-1}$ are positive definite. $\square$

## B.2    Computational Complexity

We next analyse the computational complexity of our pooling methods, which are described in Appendix C.4 and Section 3.1. Specifically, we expand on the statements in Section 3.2 by detailing the computational complexity of the pooling process. Given a graph $G = (X, E)$, let $k = \lfloor(1 - r)|X|\rfloor$ be the number of nodes that should be contracted as determined by the pooling ratio $r$. The time complexity of our pooling approach can be split up into the following steps:

**Computing magnitude or spread.** Magnitude has time complexity $O(|X|^3)$ and can further be approximated via iterative normalisation in $O(i \times |S_i| \times |X|^2)$ time assuming $G$ has a positive weighting where $i$ is the number of iterations and $S_i \subset X$ [5]. Spread computations have time complexity $O(|X|^2)$, which is a notable improvement to magnitude. It is possible to approximate spread computations via subsets [23] or iterative optimisation using mini-batching [5]. For $i$ iterations on subsets $S_i \subset X$, the time complexity of approximating spread reduces to $O(i \times |S_i| \times |X|)$ [5, 23]. Spread thus offers a much faster alternative to magnitude and can scale to large graphs considerably more efficiently.

**Computing distances and similarities.** For large datasets, it is key to speed up the distance calculations. Diffusion distances have time complexity $O(|X|^3)$, but can be reduced to $O(k|X|^2)$ when restricting the computations to the top k eigenvectors. Diffusion maps can further be approximated via low-rank approximations. To reduce the cost of repeated distance computations, it is possible to approximate the metric on the reduced graph $G/e$ by directly updating the distances for $G$. Given a distance matrix, computing the similarity matrix then has linear time complexity in the number of entries. Denote the time complexity of computing the distances and similarities by $O(C_d)$.

**Edge contraction.** To get $G' = G/e$, contracting an edge $e \in E$ takes $O(|X|)$ time.

**Edge score computations.** For each edge, its edge score is computed by applying the edge contraction and computing the magnitude or spread of the reduced graph, which takes $O(|X| + C_d + C_S)$ time, where $C_S$ refers to the cost of computing either magnitude or spread as detailed above. Note that the computation of these edge scores is independent across edges and can be parallelised.

**Edge score sorting.** The edge scores can be sorted from lowest to highest in $O(|E| \log |E|)$ time.

**Feature aggregation.** The node features, $\mathbf{F} \subseteq \mathbb{R}^{|X|,F}$, can be aggregated in $O(|X| \times f)$ time.

Putting this all together, we get that the cost of our pooling algorithms can be described by a worst-case time complexity of

$$O(|E|(|X| + C_d + C_S + log|E|) + |X|(f + k))$$

if $k \leq 0.5|X|$ and the graph is not pooled to less than half its size. Otherwise, if $k > 0.5|X|$, we re-compute the edge scores whenever no valid edges are left as described in Section 3.1. In this scenario, the first term of the complexity expression is repeated, corresponding to re-computations on successively smaller graphs. In summary, the overall time complexity of our pooling method is dominated by the cost of calculating and sorting the edge scores. This cost is independent of the choice of GNN architecture, ensuring that the training costs remain stable and do not escalate with model complexity. In practice, as further explored in Appendix D.2, we thus find that our pooling algorithms perform on par with alternative pooling layers in terms of computational efficiency.

## B.3 Magnitude and Spread

### B.3.1 Additivity for disjoint graphs

As a measure of the effective size, one appealing property of magnitude is that it behaves akin to cardinality. In fact, magnitude is additive when taking the disjoint union of multiple metric spaces.

**Theorem 2.** *Consider a graph $G = G_1 \sqcup G_2$ consisting of the disjoint union of two graphs $G_1$ and $G_2$. Then $Mag(G) = Mag(G_1) + Mag(G_2)$.*

*Proof.* Let $G = (X, E)$ together with the metric $d$ be a positive definite metric graph. Assume $G$ is a disjoint union of two metric graphs $(G_1, d_1)$ and $(G_2, d_2)$ over the vertex sets $X_1$ and $X_2$, respectively, such that $d_{|X_1} = d_1$ and $d_{|X_2} = d_2$ and $d(x_1, x_2) = \infty$ for all $x_1 \in X_1$, $x_2 \in X_2$. Then, $\zeta_G = \zeta_{G_1} \oplus \zeta_{G_2}$ and $\mathrm{Mag}(G) = \mathrm{Mag}(G_1) + \mathrm{Mag}(G_2)$. $\qquad\square$

This result applies to graphs equipped with the shortest-path distance or the diffusion distance considered in this paper, because the distance between two nodes depends only on the connected component they belong to and is infinite if there is no path between them. Therefore, we can naturally see the similarity matrix $\zeta_G$ as block-diagonal and compute the magnitude of $G$ by summing up the magnitude of its disconnected subgraphs.

### B.3.2 Isomorphism invariance

A key property of magnitude and spread is that they are isometry invariants of metric spaces. Note that by *graph isometry* we mean an isometry on the underlying vertex set equipped with a metric.

**Theorem 3.** *For isomorphic graphs $G_1, G_2$, we have $Mag(G_1) = Mag(G_2)$ and $Sp(G_1) = Sp(G_2)$.*

*Proof.* Let $f : (X_1, d_1) \to (X_2, d_2)$ be a bijective isometry between the metric graphs $G_1$ and $G_2$, respectively. Then, for all $x, y \in X_1$ we have that $d_2(f(x), f(y)) = d_1(x, y)$. A consequence of the bijectivity of $f$ is that the distance matrices coincide (up to permutations) and that $\zeta_1 = \zeta_2$. This implies in turn that $\mathrm{Mag}(G_1) = \mathrm{Mag}(G_2)$ and that $\mathrm{Sp}(G_1) = \mathrm{Sp}(G_2)$. $\qquad\square$

Based on this property for magnitude and spread, we can show that isometry invariance also holds for our proposed pooling algorithm further detailed in Appendix C.4.

**Corollary 1.** *MagEdgePool and SpreadEdgePool are isometry-invariant if applied to isomorphic graphs provided the choice of edges to contract at each iteration is deterministic whenever the edge scores coincide.*

*Proof.* Let $f : (G_1, d_1) \to (G_2, d_2)$ be a graph isomorphism and an isometry. From Theorem 3, we know that the edge scores as defined in Equation (5) for $e_1 \in E_1$ and $f(e_1) \in E_2$ will coincide, i.e. $s(e_1) = s(f(e_1))$. Because the choice of edges to contract at each iteration is further assumed to be deterministic if edge scores coincide, it follows that every edge to contract in $G_2$ corresponds to the image $f(e)$ of an edge $e$ to contract in $G_1$ and vice versa. Hence, the pooled graphs output by our algorithm are isomorphic. $\qquad\square$

### B.3.3 Edge contraction on graphs

We first recall important results about magnitude in the context of (strictly) positive definite finite metric spaces and refer the interested reader to Leinster [37] for further details.

**Proposition 1** (Proposition 2.4.3, Leinster [37]). *Let $(X, d)$ be a positive definite metric space with finite cardinality $|X| = n$. Then*

$$Mag(X) = \sup_{v \in \mathbb{R}^n \setminus \{0\}} \frac{(\sum_{i=0}^n v_i)^2}{v^t \zeta_X v}. \tag{8}$$

Proposition 1 implies that the magnitude of positive definite metric spaces is always positive. Another consequence of this result is a *monotonicity* property on subsets of these metric spaces.

**Corollary 2** (Corollary 2.4.4, Leinster [37]). *Let $(X, d)$ be a positive definite finite metric space and consider a subset $Y \subset X$ (endowed with the induced metric). Then*

$$Mag(Y) \leq Mag(X). \tag{9}$$

We will now show an analogous result for graphs constructed via edge contraction.

**Theorem 4.** *Consider an edge-contraction map $f : (G_1, d_1) \to (G_2, d_2)$ between positive definite metric graphs. If the map is 1-Lipschitz, i.e. $d_2(f(v_1), f(v_2)) \leq d_1(v_1, v_2) \ \forall v_1, v_2 \in X_1$, then $Mag(G_2) \leq Mag(G_1)$.*

*Proof.* Let $f : (G_1, d_1) \to (G_2, d_2)$ be an edge-contraction map and let $n := |X_1|$ and $m := |X_2|$. We will identify $\mathbb{R}^m$ with a subset of $\mathbb{R}^n = \mathbb{R}^m \oplus \mathbb{R}^{n-m}$ using the map $(v_1, \cdots, v_m) \in \mathbb{R}^m \hookrightarrow (v_1, \cdots, v_m, 0, \cdots, 0) \in \mathbb{R}^n$. Then,

$$v^t \zeta_{X_1} v = \sum_{i,j} v_i \zeta_{X_1}[i,j] v_j$$

$$= \sum_{i,j} v_i \big(e^{-d_1(x_i, x_j)}\big) v_j$$

$$\leq \sum_{i,j} v_i \zeta_{X_2}[i,j] v_j,$$

and

$$\frac{1}{v^t \zeta_{X_2} v} \leq \frac{1}{v^t \zeta_{X_1} v} \ \forall \ v \neq 0. \tag{10}$$

Finally, by Proposition 1 and Equation (10), we get that

$$\mathrm{Mag}(G_1) = \sup_{v \in \mathbb{R}^n \setminus \{0\}} \frac{(\sum_{i=0}^n v_i)^2}{v^t \zeta_{X_1} v}$$

$$\geq \sup_{v \in \mathbb{R}^m \setminus \{0\}} \frac{(\sum_{i=0}^n v_i)^2}{v^t \zeta_{X_1} v}$$

$$\geq \sup_{v \in \mathbb{R}^m \setminus \{0\}} \frac{(\sum_{i=0}^m v_i)^2}{v^t \zeta_{X_2} v}$$

$$= \mathrm{Mag}(G_2)$$

$\square$

### B.3.4 Bounding magnitude by spread

Recall that for positive definite metric spaces, magnitude is known to be an upper bound for spread.

**Theorem 6** (Theorem 2.2. from Willerton [52]). *Suppose that $X$ is a finite metric space. If $X$ is positive definite then*

$$Sp(X) \leq Mag(X).$$

We will now use this bound as well as the results in Appendix B.3.3 to investigate the relationship between MagEdgePool and SpreadEdgePool. Through a process of iterated edge contraction, our pooling algorithm produces a sequence of hierarchically pooled graphs (as described in Section 3.1 and Appendix C.4). Note that it is not guaranteed that MagEdgePool and SpreadEdgePool yield the same sequence. For this reason, we will refer to the graphs resulting from the $k^{th}$ edge-contraction with MagEdgePool and SpreadEdgePool by $G^{(k)}$ and $\widetilde{G}^{(k)}$ respectively.

For each $k$, the edge contraction map $G^{(k)} \to G^{(k+1)}$ is a surjection on the underlying vertex sets $X^{(k)}$ and $X^{(k+1)}$ respectively, i.e $X^{(k+1)} \subset X^{(k)}$. Moreover, we will assume that for any $k$ this map is distance-decreasing. That is, $d^{(k+1)}(f(x_i), f(x_j)) \leq d^{(k)}(x_i, x_j)$ for all $x_i, x_j \in X^{(k+1)}$.

Recall that for any $k$, $\text{Mag}(G^{(k)})$ is the magnitude of a finite positive definite metric space $(X^{(k)}, d^{(k)})$. Then, by Theorem 4, we deduce that for any $k$,

$$\text{Mag}(G^{(k+1)}) \leq \text{Mag}(G^{(k)}) \tag{11}$$

Note that by construction, scoring the edges in Algorithm 1 translates into the following:

$$X^{(k)} = \text{argmin}_{Y \subset X^{(k-1)}, |Y|+1=|X^{(k-1)}|} |\text{Mag}(X^{(k-1)}) - \text{Mag}(Y)| \tag{12}$$

and,

$$\widetilde{X}^{(k)} = \text{argmin}_{Y \subset \widetilde{X}^{(k-1)}, |Y|+1=|\widetilde{X}^{(k-1)}|} |\text{Sp}(\widetilde{X}^{(k-1)}) - \text{Sp}(Y)|. \tag{13}$$

Then, by the monotonicity of magnitude, i.e. Equation (11),

$$X^{(k)} = \text{argmax}_{Y \subset X^{(k-1)}, |Y|+1=|X^{(k-1)}|} \text{Mag}(Y) = \text{argmax}_{Y \subset X^{(k-1)}} \text{Mag}(Y).$$

Let $\Delta^{(k)}\text{Mag}(G) = |\text{Mag}(G^{(k-1)}) - \text{Mag}(G^{(k)})|$ and let $\Delta^{(k)}\text{Sp}(G) = |\text{Sp}(G^{(k-1)}) - \text{Sp}(G^{(k)})|$. For the following result, we assume that scores are only computed once and that they are the only criterion for edge contraction. Furthermore, we will assume that spread is monotonically decreasing.

**Theorem 5.** *Consider a positive definite metric graph $G$ with positive weights. Assume that the edge-contraction maps describing MagEdgePool and SpreadEdgePool induce distance decreasing surjections on the vertex sets. If $|\text{Mag}(G^{(k-1)}) - \text{Sp}(G^{(k)})| \leq C\Delta^{(k)}\text{Sp}(G)$, then*

$$\Delta^{(k)}Mag(G) \leq 3C\Delta^{(k)}Sp(G).$$

*Proof.* For any $k$ we have the following inequality:

$$|\text{Mag}(G^{(k-1)}) - \text{Mag}(G^{(k)})| \leq |\text{Sp}(G^{(k-1)}) - \text{Sp}(G^{(k)})| + |\text{Mag}(G^{(k)}) - \text{Sp}(G^{(k)})|$$
$$+ |\text{Mag}(G^{(k-1)}) - \text{Sp}(G^{(k-1)})|.$$

Assume that $|\text{Mag}(G^{(k-1)}) - \text{Sp}(G^{(k)})| \leq C\Delta^{(k)}\text{Sp}(G)$ for some constant $C > 0$. By the monotonicity of magnitude (Theorem 4), we get that $\text{Mag}(G^{(k)}) \leq \text{Mag}(G^{(k-1)})$ and,

$$\text{Mag}(G^{(k)}) - \text{Sp}(G^{(k)}) \leq \text{Mag}(G^{(k-1)}) - \text{Sp}(G^{(k)}) \leq C\Delta^{(k)}\text{Sp}(G).$$

Similarly, assuming monotonicity of spread yields $\text{Sp}(G^{(k)}) \leq \text{Sp}(G^{(k-1)})$ and,

$$\text{Mag}(G^{(k-1)}) - \text{Sp}(G^{(k-1)}) \leq \text{Mag}(G^{(k-1)}) - \text{Sp}(G^{(k)}) \leq C\Delta^{(k)}\text{Sp}(G).$$

Since $0 \leq \text{Sp}(G^{(k-1)}) - \text{Sp}(G^{(k)}) \leq \text{Mag}(G^{(k-1)}) - \text{Sp}(G^{(k)})$, the constant $C$ must be greater than or equal to 1. We conclude that

$$\Delta^{(k)}\text{Mag}(G) \leq 3C\Delta^{(k)}\text{Sp}(G).$$

$\square$

### B.4  Expressivity

While studying the expressive power of GNNs, we aim to evaluate their ability to generate different outputs for non-isomorphic graphs. Here, we will analyse the expressive power of our pooling methods MagEdgePool and SpreadEdgePool within the *Select-Reduce-Connect* framework introduced by Grattarola et al. [32] for describing pooling operators. Let $G = (X, E)$ be a graph and $\mathbf{F} \in \mathbb{R}^{|X| \times f}$ be the node features associated to the nodes in $G$. Then, a graph pooling operator is regarded as a function **POOL**: $(\mathbf{F}, G) \mapsto (\mathbf{F}_P, G_P)$ where $\mathbf{F}_P$ denotes the pooled node features and $G_P = (X_P, E_P)$ the pooled graph with $|X_P| \leq |X|$. Pooling is described as a combination of three elementary functions: *selection* (**SEL**), *reduction* (**RED**), and *connection* (**CON**). The selection function clusters the nodes of the input graph into super-nodes, so that **SEL**: $G \mapsto \mathcal{S} = \{\mathcal{S}_j\}_{j=1}^{|X_P|}$ where $\mathcal{S}_j = \{S_{ij}\}_{i=1}^{|X|}$ and $S_{ij}$ is the membership score of the node $i$ to super-node $j$. Node selection can be represented as the assignment matrix $S \in \mathbb{R}^{|X| \times |X_P|}$ with entries $S_{ij}$. Based on this selection, the reduction function aggregates node features of all nodes that are assigned to the same super-node, i.e. **RED**: $(\mathbf{F}, S) \mapsto \mathbf{F}_P$. Then, the connection function, **CON**, generates the edges and determines the connectivities between super-nodes in the pooled graph. Finally, to study expressivity, we note that hierarchical graph pooling is typically applied in GNN architectures after some initial message passing layers. Let $G^L$ denote the graph resulting from a block of $L$ MP layers and $\mathbf{F}^L \in \mathbb{R}^{|X| \times f}$ the corresponding feature matrix [8].

**Theorem 7.** *(Theorem 1 from Bianchi and Lachi [8]) Let $G_1 = (X_1, E_1)$ and $G_2 = (X_2, E_2)$ be two graphs equipped with node features $\mathbf{F}_i \in \mathbb{R}^{|X_i| \times f}$ for $i = 1, 2$. Assume that $G_1 \neq_{WL} G_2$, i.e. that $G_1$ and $G_2$ are distinguishable by the Weisfeiler-Leman isomorphism test. Apply a block of $L$ MP layers to get $G_1^L$ and $G_2^L$ as well as $\mathbf{F}_1^L$ and $\mathbf{F}_2^L$. Let **POOL** be a pooling operator placed after these MP layers to get $G_{1_P} = POOL(G_1^L)$ and $G_{2_P} = POOL(G_2^L)$ associated with the node features $\mathbf{F}_{1_P}$ and $\mathbf{F}_{2_P}$ in $\mathbb{R}^{k \times f}$. Then, $G_{1_P}$ and $G_{2_P}$ will have different node features (up to permutation) provided the following conditions hold:*

1. *$\sum_{i=1}^{|X_1|} \mathbf{F}_{1_{[i,:]}}^L \neq \sum_{i=1}^{|X_2|} \mathbf{F}_{2_{[i,:]}}^L$,*
2. *The memberships generated by **SEL** satisfy $\sum_{j=1}^{k} S_{ij} = \lambda$, with $\lambda > 0$ for each node $i$, i.e., the cluster assignment matrix $S$ is a right stochastic matrix up to the global constant $\lambda$,*
3. *The reduction function satisfies **RED**: $(\mathbf{F}^L, S) \mapsto \mathbf{F}_P = S^T \mathbf{F}^L$.*

The WL test will identify that two graphs with different multisets of node features are non-isomorphic based on the injectivity of the colouring function of the WL algorithm. Theorem 7 then guarantees that $G_{1_P} \neq_{WL} G_{2_P}$ thus ensuring that the pooling operation **POOL** preserves expressivity.

**Corollary 3.** *MagEdgePool and SpreadEdgePool satisfy the sufficient conditions outlined in Theorem 7 when using sum aggregation for pooling the node features.*

*Proof.* Condition 1 is independent to the choice of pooling layer and instead relates to the expressivity of MP layers. It is guaranteed for any MP layer that is as powerful as the 1-WL test [8, 25]. Conditions 2 and 3 hold trivially by construction. Each super-node is the result of edge contractions and node features are aggregated via summation. Hence, each vertex is assigned to a unique super-node and the selection matrix is constructed as $S_{ij} = 1$ if the node $i$ is contained in super-node $j$ and $S_{ij} = 0$ otherwise. This ensures that $\sum_{j=1}^{k} S_{ij} = 1$ for every node $i$ fulfilling condition 2. The resulting **SEL** function can be represented using the *cluster assignment matrix $S$* obtained as a product of these elementary (contraction) operations represented by matrices. This yields the **RED** function described in our algorithm in Algorithm 1 as a map $\mathbf{F}^L \mapsto S^T \mathbf{F}^L$ that respects condition 3. $\qquad\square$

## C Extended Methods

### C.1 Hardware and Software

The experiments reported in our study were implemented using `spektral 1.3.1` [31][2], and `tensorflow 2.16.2` [1][3]. As further detailed in Appendix C.5.3 and Section 4.1, we base our graph classification experiments on the benchmark setup and code by Grattarola et al. [32][4], which also include implementations for the pooling layers compared across our study. By relying on this existing framework, we aim to ensure the reproducibility of our results.

Further, to calculate magnitude and spread we rely on `magnipy`, a Python package by Limbeck et al. [40] for magnitude and diversity computations.[5] Further, as a novel contribution of this paper, we extend the computation of magnitude to graph data and novel graph metrics. Specifically, we modify the computations, so that the magnitude of disconnected subgraphs is computed separately (based on Theorem 2) using the `NetworkX`[6] package. We also implement graph distances that have not previously been used to compute magnitude, such as the diffusion distances detailed in Section 2.3. Further details on the code for implementing our proposed pooling methods and our experiments can be found in our supplementary code submission as well as on GitHub. Finally, we publish a reproducible `PyTorch` implementation of our pooling methods as `mag_edge_pool`[7] on GitHub.

All experiments were conducted on a high-performance cluster with hardware specifications as detailed in Table S.1. In particular, all experiment were run requesting a single GPU with 32 GB video memory or less.

Table S.1: Summary of the compute resources used for our experiments.

| Inventory | Models |
|---|---|
| Available CPUs | Intel Xeon (Gold 6128, 6130, 6134, 6136, 6142, 6240, 6248R) |
| | Intel Xeon Platinum (8280L, 8480+, 8468, 8562Y+) |
| | Intel Xeon (E7-4850, E5620, 4114, 6126) |
| | AMD EPYC (7262, 7413, 7513, 7713, 7742) |
| | AMD Opteron (6128, 6164 HE, 6234, 6272, 6376 x2) |
| Available GPUs | NVIDIA Tesla (K80, P100, V100) |
| | NVIDIA A100 (20GB, 40GB, 80GB PCIe) |
| | NVIDIA H100 (80GB PCIe) |
| | NVIDIA Quadro RTX 8000 |

### C.2 Datasets

We briefly describe the graph datasets analysed throughout our work. Simulated graphs, as used for Figure 1, are created using either `PyGSP` [8] or `NetworkX`[9] and all example graphs were created to consist of 64 nodes.

For our main graph classification experiments, we analyse six graph datasets taken from biological or chemical applications [11, 22, 46, 47, 49], and two datasets which represent social networks [54]. All datasets are taken either from the `TUDataset`[10] benchmark [45] or the Open Graph Benchmark[11].

---

[2] https://graphneural.network/ available under an MIT license.

[3] https://pypi.org/project/tensorflow/2.16.2/ available under the Apache Software License (Apache 2.0).

[4] https://github.com/danielegrattarola/SRC available to the research community (Grattarola et al. [32]).

[5] https://github.com/aidos-lab/magnipy available under a BSD 3-Clause License.

[6] https://github.com/networkx/networkx available under a BSD 3-Clause License.

[7] https://github.com/aidos-lab/mag_edge_pool available under a BSD 3-Clause License.

[8] https://pygsp.readthedocs.io/en/stable/ available under a BSD-3-Clause license.

[9] https://github.com/networkx/networkx available under a BSD 3-Clause License.

[10] https://chrsmrrs.github.io/datasets/ available under a CC-BY-4.0 license.

[11] https://ogb.stanford.edu/ available under an MIT licence.

More specifically, the results in Table 1, Table 2, and Table S.8 analyse the following graph datasets described in Table S.2. Note that we only consider node and not edge features for our experiments.

Table S.2: Summary of the graph datasets considered for our experiments.

| dataset | library | # classes | # node features | # graphs | avg # nodes | avg # edges |
|---|---|---|---|---|---|---|
| **MUTAG** | TUDataset | 2 | no | 187 | 18 | 40 |
| **Enzymes** | TUDataset | 6 | 18 | 600 | 33 | 62 |
| **COX2** | TUDataset | 2 | 3 | 467 | 41 | 44 |
| **DHFR** | TUDataset | 2 | 3 | 756 | 42 | 45 |
| **IMDB-B** | TUDataset | 2 | no | 1000 | 20 | 97 |
| **IMDB-M** | TUDataset | 3 | no | 1500 | 13 | 65 |
| **AIDS** | TUDataset | 2 | 4 | 2000 | 15 | 16 |
| **Proteins** | TUDataset | 2 | 29 | 1113 | 39 | 72 |
| **Mutagenicity** | TUDataset | 2 | no | 4337 | 30 | 31 |
| **NCI1** | TUDataset | 2 | no | 4110 | 30 | 32 |
| **NCI109** | TUDataset | 2 | no | 4127 | 30 | 32 |
| **OGBG-MOLHIV** | OGB | 2 | 9 | 41127 | 25 | 27 |
| **BZR** | TUDataset | 2 | 3 | 405 | 36 | 38 |
| **BZR_MD** | TUDataset | 2 | no | 306 | 21 | 225 |
| **COX2_MD** | TUDataset | 2 | no | 303 | 26 | 335 |
| **DHFR_MD** | TUDataset | 2 | no | 393 | 24 | 283 |
| **ER_MD** | TUDataset | 2 | no | 446 | 21 | 235 |
| **OGBG-MOlESOL** | OGB | regression | 9 | 1128 | 13 | 14 |
| **OGBG-MOlFREESOLV** | OGB | regression | 9 | 642 | 9 | 8 |
| **OGBG-MOLLIPO** | OGB | regression | 9 | 4200 | 27 | 30 |

## C.3 Magnitude and Spread Computations

Across our experiments, we compute magnitude and spread as outlined in the main text, implemented in our code submission, and further described in Appendix C.1. Elaborating on these descriptions, we now aim to give an extended explanation of practical and theoretical considerations for computing the magnitude of graphs in practice.

**Defining the magnitude of a graph.** In mathematical literature, the magnitude of graphs is often studied with the shortest path metric [38]. However, shortest path distances are not guaranteed to be of negative type, thus leading to scenarios and well-known examples for which the similarity matrix is not invertible and magnitude based on this metric cannot be computed [37]. In comparison, resistance distances, diffusion distances, or Euclidean distance always permit the computation of magnitude. Because of this difference in the choice of distance metric, we note that our definition of the magnitude of a graph in Section 2.2 differs from the definition used by Leinster [38]. While we choose to investigate diffusion distances, we note that the distance metric can easily be replaced if needed to explore alternative geometries.

**Diffusion geometry.** For further research, we believe that the usage of diffusion distances offers the chance to leverage a rich theory on approximation methods via landmarks [42], or localised diffusion computations [16], which can lead to further computational improvements and extension of our methods.

**Magnitude and spread as multi-scale functions.** Note that magnitude and spread can also be defined as multi-scale functions i.e. $t \mapsto \text{Mag}((X, t \cdot d))$ for a metric space $(X, d)$ and a scale parameter $t \in \mathbb{R}^+$. This parameter $t$ can be likened to choosing a kernel bandwidth or the scale of distances or similarity determining when observations are considered to be distinct. In practical applications, it is advisable to carefully consider the choice of scaling factor $t$ or the type of normalisation used to compare distances [40]. Limbeck et al. [40] propose a heuristic that uses root-finding to find a suitably large $t$. However, this requires repeated computations of magnitude and increases the computational costs. A faster and more desirable default choice of $t$ would be based solely on the distance metric. For distances that are not otherwise scaled or normalised, we therefore recommend the usage of faster heuristics, such as the median heuristic for choosing the kernel-bandwidth i.e. the scale parameter $t$ [29]. For diffusion distances, we find that setting $t = 1$ is sufficient for our goals. This is because, as discussed in Section 2.3, diffusion distances are computed from the normalised

graph Laplacians and are inherently comparable across graphs. Nevertheless, investigating magnitude and spread as multi-scale functions on graphs remains an interesting extension for further work.

**Magnitude and spread as diversity measures.** We extensively discuss the relationship between magnitude and spread throughout our work. However, our main paper does not have the space to fully explain the theoretical motivations behind the formulations of magnitude, spread, and other generalised measures of diversity. For a more complete discussion we refer the interested reader to Leinster [39], an extensive reference work on the mathematical motivation behind entropy and diversity. Furthermore, Willerton [52] specifically discusses the spread of a metric space, and Limbeck et al. [40] describe the practical usage of magnitude as a diversity measure in ML. These works also give descriptions on how and why the magnitude or spread of a metric space can be interpreted as an effective size i.e. as the effective number of distinct points in a metric space or the number of dissimilar nodes in a graph.

## C.4 Pooling Algorithm

We now detail our pooling algorithm introduced in Section 3.1 by describing a pseudocode implementation. Note that to describe the algorithm we assume we have pre-selected a distance metric for computing either magnitude or spread.

---

**Algorithm 1** Graph Pooling Methods: SpreadEdgePool and MagEdgePool

---

**Require:** input graph $G = (X, E)$, node features $\mathbf{F} \in \mathbb{R}^{|X| \times f}$, pooling ratio $r \in (0, 1]$, diversity measure $\mathrm{Mag}(G)$ or $\mathrm{Sp}(G)$
**Ensure:** Pooled graph $G' = (X', E')$, pooled features $\mathbf{F}'$
  1: Initialise the super-node set $\mathcal{S}(x) \leftarrow x$ for all $x \in X$
  2: Initialise the set of edges adjacent to a contracted edge $E_c \leftarrow \emptyset$
  3: Initialise the pooled graph $G' \leftarrow G$
  4: Compute initial edge scores:

$$s(e) = |\mathrm{Mag}(G) - \mathrm{Mag}(G/e)| \quad \forall e \in E$$

  5: **while** $|X'| \neq \lfloor r|X| \rceil$ AND $|E'| \neq \emptyset$ **do**
  6:     Select edge $e = (x, y) = \arg\min_{e \in E \setminus E_c} s(e)$
  7:     **if** $e$ is not adjacent to any previously contracted edge in $E_c$ **then**
  8:         Contract edge $e$, update $G' \leftarrow G'/e$
  9:         Add $e$ and any edges adjacent to $e$ to $E_c$
10:         Update the node selection: merge $\mathcal{S}(x)$ and $\mathcal{S}(y)$
11:     **end if**
12:     **if** no more valid edges AND pooling ratio not reached **then**
13:         Recompute the edge scores $s(e)$ on the updated graph $G'$
14:         Reset $E_c \leftarrow \emptyset$
15:     **end if**
16: **end while**
17: Initialize $\mathbf{F}' \leftarrow \emptyset$
18: **for** each super-node representative $w \in \mathcal{S}$ **do**
19:     Let $S_w = \{x \in X \mid \mathcal{S}(x) = w\}$
20:     Compute the aggregated features:

$$\mathbf{F}'_{w,:} = \frac{1}{|S_w|} \sum_{x \in S_w} \mathbf{F}_{x,:}$$

21:     Append $\mathbf{F}'_w$ to $\mathbf{F}'$
22: **end for**
23: **return** Pooled graph $G'$, pooled features $\mathbf{F}'$

---

## C.5 Extended Experimental Details

We briefly describe extended details on the experimental setup used for our main experiments.

### C.5.1 Overview Experiment

To create the illustration in Figure 1 and visually compare the outputs of different pooling methods, we follow the experimental setup by Grattarola et al. [32] on understanding structure preservation in graph pooling layers. Specifically, we simulate a ring graph with 64 nodes, a barbell graph with 20 nodes on each side connected by 24 nodes in the middle, and a sensor graph with 64 nodes. Graphs are then pooled to a pooling ratio of approximately $50\%$. Because some pooling methods (e.g. Graclus or NDP) do not give exact control over the number of vertices, but pool graphs to approximately half of their original size, the number of nodes visualised in Figure 1 can vary across methods. All trainable pooling layers were then trained in a self-supervised manner to optimise the following spectral loss between the original graph $G$ and the pooled graph $G'$:

$$\mathcal{L}(G, G') = \sum_{i=0}^{f} F_{:,i}^{\top} L F_{:,i} - F_{:,i}'^{\top} L' F_{:,i}' \tag{14}$$

where $L, L'$ are the corresponding Laplacian matrices. The features $F$ are taken to be the top 10 eigenvectors of $L$ concatenated with the coordinates of the nodes in $G$. $F'$ is the reduced version of $F$ after pooling. Note that this type of spectral loss is one particular proposal on structure preservation and alternative objectives could be investigated.

### C.5.2 Structure Preservation Experiment and Pooling Ratios

For reporting the structure preservation results described in Section 4.2, we considered multiple different proposals for what it means to preserve graph structure during pooling. In the end, we settled to compare the spectral distance between the symmetrically normalised graph Laplacians as an established measure of spectral property preservation, and investigated the relative difference in magnitude between the original graph $G$ and the pooled graph $G'$ computed from diffusion distances. Specifically, the reported relative magnitude difference is calculated as

$$\text{MagDiff}(G, G') = \frac{|\text{Mag}(G) - \text{Mag}(G')|}{\text{Mag(G)}}. \tag{15}$$

For this experiment we further vary the pooling ratios across different pooling methods. However, some of the pooling layers considered in our study (NDP, Graclus and NMF) were configured to always pool graphs to around half their size. To allow us to compare these methods across increasing pooling ratios, we choose to reapply these pooling operations repeatedly, which is why these three pooling methods are evaluated at pooling ratios that are powers of 0.5.

### C.5.3 Graph Classification Experiment

Our graph classification architecture follows the experimental setup described in Section 4.1 and is based the benchmark by Grattarola et al. [32]. Across our main classification experiment detailed in Section 4.1, different pooling layers are configured to reduce each input graph to around 50% of the number of nodes in the original graphs. Depending on the pooling method, this is chosen so each graph is reduced to 50% of its original size $k = \lfloor 0.5 * N \rfloor$, (for NDP, Graclus, MagEdgePool, SpreadEdgePool, TopKPool, and SAGPool), or to 50% the average size of all graphs in the training dataset $k = \lfloor 0.5 * \bar{N} \rfloor$ (for DiffPool, and MinCutPool). Interpreting the experimental results in Table 1 it is thus of interest that the sizes of the pooled graphs can vary across pooling layers, which might explain some of the difference in performance between the fixed-size methods DiffPool and MinCUT compared to more adaptive pooling methods.

### C.5.4 Graph Regression

As a further ablation study, we aim to assess whether the results reported in Table 1 for graph classification tasks remain consistent for further graph regression tasks. To this end, we use three molecular datasets from the OGB benchmark with their predefined test, training, and validation

splits [33]. Further, we adjust the GNN architecture described in Section 4.1 and Appendix C.5.3 to use the MSE as a training loss, the RMSE for performance evaluation, a linear final activation for the readout MLP, an early stopping patience of 100 epochs, and blocks of two convolutional layers instead of single layers. Similar to before, we only use node features as inputs. Note that stronger, domain-specific, and purpose-built models exist for molecular regression tasks that utilise both atom and bond information [33]. Hence, our goal is not to reach state of the art performance. Rather, we aim to compare the performance of different pooling layers and evaluate the information loss due to the pooling operations themselves. Finally, Table 2 reports the RMSE on the test dataset across ten repeats using varying random seeds.

## D  Extended Results

Finally, we summarise extended experimental results beyond the scope of our main paper.

### D.1  Correlation between Magnitude and Spread

As stated in Section 2.2, the magnitude and spread of a metric space are closely related with magnitude giving an upper bound for spread when computed from the same positive definite metric space. Across our experiments on real graph datasets, we further find that this bound in practice can be very tight and magnitude and spread measure very related notions of effective size. More specifically, when computing both magnitude and spread from the diffusion distances detailed in Section 2.3, we observe that magnitude and spread almost coincide for all graphs from the NCI1, ENZYME or IMDB-Multi datasets as illustrated in Figure S.1. In fact, magnitude and spread correlate almost perfectly across these three graph datasets (Pearson correlation $r^2 \geq 0.99$). Further, we confirm that across these examples, magnitude is generally greater or equal to spread by a relatively low multiplicative factor close to 1. We therefore find empirical evidence for the fact that spread offers a valid and highly related alternative to magnitude in practice supporting our theoretical analysis of the relationship between spread and magnitude during pooling (Appendix B.3) as well as our observations on the similar performance of MagEdgePool and SpreadEdgePool.

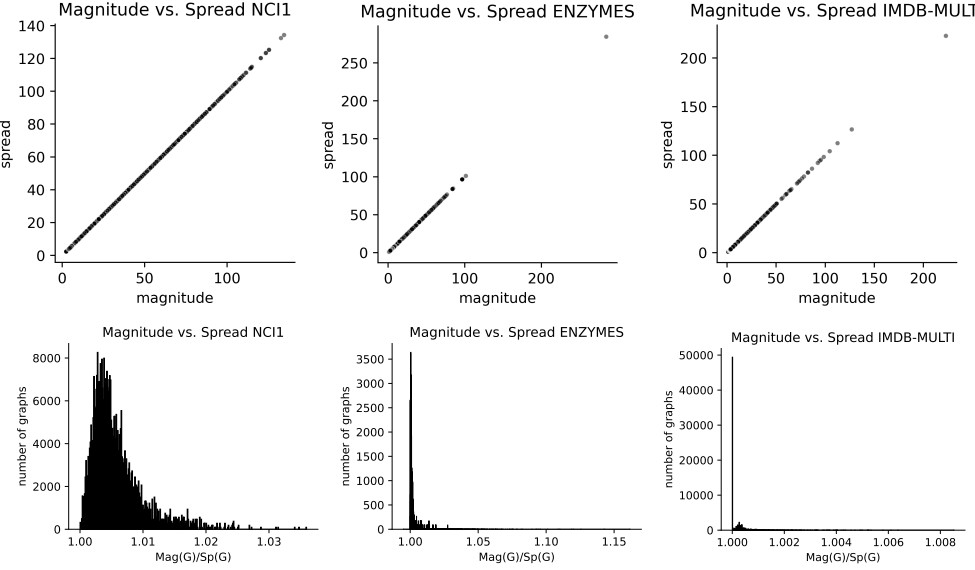

Figure S.1: Comparison between magnitude and spread computed from diffusion distances for all graphs in three graph datasets, NCI1, ENZYMES and IMDB-Multi.

## D.2 Evaluating Computational Efficiency

As detailed in Appendix B.2, the computational costs of our algorithm are determined by the costs of computing the edge scores used for pooling. To illustrate how this theoretical discussion translates into practice, we now investigate computational costs empirically in comparison to alternative pooling methods considered throughout this study.

### D.2.1 Training Costs

We first compare the runtime (in seconds) and memory usage (in MB per cross-validation run) of our pooling methods (MagEdgePool and SpreadEdgePool) to trainable pooling methods in Table S.3. Specifically, we train the GNN architecture specified in Appendix C.5.3 and Section 4.1 using GIN layers across 200 epochs using 10-fold stratified cross-validation. We compare our edge pooling methods (MagEdgePool, SpreadEdgePool) to trainable pooling methods from `torch_geometric`[12] (EdgePool, TopKPool, SAGPool) or from `torch-geometric-pool`[13] (DiffPool, MinCutPool). We record the mean and standard deviation of the runtimes in seconds in Table S.3 and GPU memory usage in MB per cross-validation run in Table S.4. We note, in particular, that our proposed edge pooling methods, MagEdgePool and SpreadEdgePool, allow for significantly more efficient GNN training than EdgePool as highlighted in Table S.3. Similarly, our methods generally improve on the runtimes for dense pooling methods such as DiffPool and MinCutPool. To generalise this runtime comparison to other datasets sizes and experimental setups, we note that our algorithm will scale with dataset size as described in Appendix B.2.

### D.2.2 Pre-training Costs

Having observed that pre-computed edge-pooling speeds up GNN training times, we further investigate the computational costs of this preprocessing step by comparing the runtime and memory costs of our methods against other non-trainable pooling operators (NDP, NMF and Graclus) prior to training. Table S.5 and Table S.6 report the computational costs of computing the pooling assignment for all graphs in the datasets. We observe that SpreadEdgePool is generally more efficient than MagEdgePool. Beyond exact computations of our pooling methods (described in Appendix C.4), Table S.5 and Table S.6 also present approximate versions that reduce the cost of distance computations. Specifically, these approximate versions, referred to as *MagEdgePool\** and *SpreadEdgePool\**, use the minimum distance to the original nodes to update the (diffusion) distances during edge contraction. This leads to a considerable improvement in runtime and memory usage during pre-training. The results reported in this section highlight one of the limitations of our edge pooling approach, namely, it scales in the number of edges as well as in the number of nodes as detailed in Appendix B.2. Nevertheless, our proposed pooling methods still outperform EdgePool in terms of computational efficiency when considering both pre-processing and training costs.

Furthermore, the pre-training costs of our method, range in a number of seconds, need to be computed only once per dataset and the memory requirements remain below what is required by GNN training. Hence, while the scalability to very large graphs is a limitation, we find that our proposed pooling methods scale sufficiently well to standard graph datasets. In practice, based on the computational complexity (Appendix B.2), we recommend that our pooling method is particularly suitable for small to medium graphs that show a certain degree of sparsity rather than being fully connected. As visualised in Appendix D.2.2, graphs with up to a few hundred nodes can feasibly be processed with our method in a matter of seconds. For future work, we believe that there is a strong potential for adapting our methods to scale on large graphs. For instance, edge score calculations could be parallised, sampling heuristics could restrict the edge score computations to a subset of candidate edges, or edge scores could be estimated from local subgraphs to improve the computations.

### D.2.3 Runtimes across Pooling Ratios

Further, we compare the runtimes of training the models used in Section 4.3 to compare the accuracy of different pooling methods across varying pooling ratios. Appendix D.2.3 then reports the mean runtime of training the GNN on one CV-fold for different choices of pooling ratios and pooling

---

[12]https://pytorch-geometric.readthedocs.io/en/latest/ available under an MIT license [26, 27].

[13]https://github.com/tgp-team/torch-geometric-pool available under an MIT license.

Table S.3: Training times in seconds compared across pooling methods. The fastest methods are marked in bold.

| Method | DHFR | ENZYMES | NCI109 | Mutagenicity | IMDB-BINARY | IMDB-MULTI |
|---|---|---|---|---|---|---|
| **MagEdge** | 39.8 ± 6.0 | **22.0 ± 0.2** | 186.0 ± 9.1 | **180.3 ± 29.0** | 75.4 ± 1.4 | **89.6 ± 6.1** |
| **SpreadEdge** | 34.8 ± 1.0 | 23.0 ± 0.7 | **181.3 ± 29.4** | 206.7 ± 20.6 | **62.7 ± 0.9** | 99.7 ± 4.3 |
| **EdgePool** | 184.1 ± 4.0 | 209.3 ± 4.1 | 807.4 ± 22.2 | 790.7 ± 22.1 | 362.3 ± 3.4 | 392.7 ± 11.9 |
| **TopKPool** | 50.7 ± 0.8 | 24.1 ± 0.6 | 226.1 ± 21.3 | 243.8 ± 15.4 | 88.2 ± 1.1 | 110.9 ± 2.2 |
| **SAGPool** | 54.4 ± 1.3 | 25.7 ± 0.4 | 250.4 ± 25.7 | 253.0 ± 20.0 | 92.6 ± 2.1 | 131.1 ± 1.3 |
| **DiffPool** | 52.2 ± 6.8 | 36.4 ± 3.3 | 242.4 ± 12.5 | 274.0 ± 21.5 | 106.7 ± 5.5 | **74.9 ± 2.0** |
| **MinCut** | **33.8 ± 2.5** | 28.2 ± 1.7 | 210.3 ± 34.4 | 201.0 ± 3.7 | 93.2 ± 1.6 | 132.8 ± 3.2 |

Table S.4: Memory usage in MB compared across pooling methods. The most efficient methods are marked in bold.

| Method | DHFR | ENZYMES | NCI109 | Mutagenicity | IMDB-BINARY | IMDB-MULTI |
|---|---|---|---|---|---|---|
| **MagEdge** | **84.0 ± 0.1** | **91.2 ± 2.5** | **97.0 ± 1.1** | **94.8 ± 1.0** | **283.4 ± 28.0** | **197.6 ± 17.8** |
| **SpreadEdge** | **84.0 ± 0.1** | **90.6 ± 2.3** | **96.8 ± 1.0** | **95.2 ± 1.0** | **283.4 ± 28.0** | **197.4 ± 17.8** |
| **EdgePool** | 125.6 ± 10.4 | 148.4 ± 15.0 | 118.6 ± 7.7 | 118.0 ± 7.4 | 666.6 ± 106.6 | 526.2 ± 89.2 |
| **TopKPool** | 89.6 ± 9.0 | 94.4 ± 4.0 | 105.0 ± 1.4 | 105.8 ± 6.5 | **259.4 ± 31.5** | **193.8 ± 17.5** |
| **SAGPool** | 104.2 ± 0.6 | 94.2 ± 3.8 | 105.2 ± 1.4 | 107.8 ± 8.1 | 273.4 ± 30.1 | 209.0 ± 17.9 |
| **DiffPool** | 90.2 ± 10.2 | 94.6 ± 2.5 | 103.8 ± 1.1 | 299.8 ± 31.9 | **258.0 ± 40.3** | 240.8 ± 26.4 |
| **MinCut** | 90.6 ± 10.3 | 94.2 ± 2.4 | 103.6 ± 1.0 | 288.4 ± 38.3 | **258.0 ± 40.3** | **196.6 ± 26.0** |

Table S.5: Pre-training times in seconds compared across non-trainable pooling methods. The fastest method is marked in bold. The fastest approximation of our pooling method is marked in italics.

| Method | DHFR | ENZYMES | NCI109 | Mutagenicity | IMDB-BINARY | IMDB-MULTI |
|---|---|---|---|---|---|---|
| **MagEdge** | 61.9 | 197.4 | 4765.1 | 678.8 | 801.8 | 480.2 |
| *MagEdge** | 24.6 | 21.0 | 81.3 | 74.9 | 77.4 | 69.9 |
| **SpreadEdge** | 66.7 | 68.3 | 316.1 | 208.8 | 287.2 | 243.8 |
| *SpreadEdge** | 10.9 | 16.0 | 52.3 | 55.6 | 56.0 | 66.0 |
| **NDP** | 5.6 | 4.1 | **37.1** | 32.6 | 5.8 | 7.0 |
| **NMF** | 7.8 | 10.8 | 39.0 | 32.1 | 8.7 | 9.0 |
| **Graclus** | **3.8** | **2.9** | 54.5 | **18.2** | **4.5** | **6.0** |

Table S.6: Pre-training memory usage in MB compared across non-trainable pooling methods. The most efficient method is marked in bold. The most efficient approximation of our pooling method is marked in italics.

| Method | DHFR | ENZYMES | NCI109 | Mutagenicity | IMDB-BINARY | IMDB-MULTI |
|---|---|---|---|---|---|---|
| **MagEdgePool** | 76.2 | 163.8 | 158.9 | 179.0 | 162.1 | 93.1 |
| *MagEdgePool** | 83.7 | *34.8* | *47.0* | *72.2* | *11.5* | 65.2 |
| **SpreadEdgePool** | 61.9 | 60.1 | 177.9 | 93.1 | 204.1 | 75.8 |
| *SpreadEdgePool** | *33.6* | 54.1 | 60.5 | 72.9 | 53.2 | *23.8* |
| **NDP** | **3.0** | **5.0** | **7.0** | 13.4 | 5.1 | 6.5 |
| **NMF** | 3.8 | 5.9 | 39.9 | **10.5** | 4.2 | 5.9 |
| **Graclus** | 15.8 | 7.5 | 17.5 | 59.5 | **1.0** | **1.5** |

methods for the NCI1 dataset using either general convolutional layers or GIN layers. All models are trained as specified in Section 4.1 on a single GPU with 32 GB memory. Notably, we observe that MagEdgePool and SpreadEdgePool overall perform on par with alternative pooling methods in terms of runtimes. SpreadEdgePool has a consistent advantage over MagEdgePool due to the higher computational efficiency of computing spread rather than magnitude. Notice that for increasing pooling ratios, our algorithm re-computes the edge scores repeatedly, leading to a less pronounced decrease in computational costs than alternative methods. Nevertheless, we conclude that it is generally more efficient to apply SpreadEdgePool than to rely on trainable approaches, such as TopK and SAGPool for this dataset, indicating the computational benefit of non-trainable graph pooling operations.

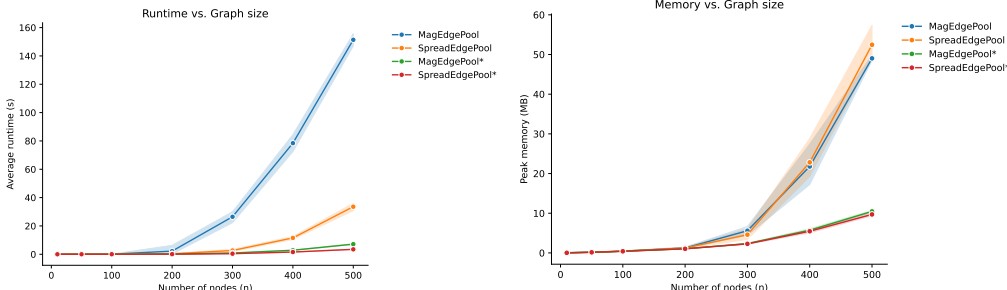

Figure S.2: Runtime and memory costs of computing the pooling assignment for Erdős-Rényi graphs with edge probability $0.005$ for increasing numbers of nodes. Lines show the mean and shaded areas the standard deviation across five repeats. SpreadEdgePool is notably faster than MagEdgePool. Distance approximations (marked with *) further speed up computations.

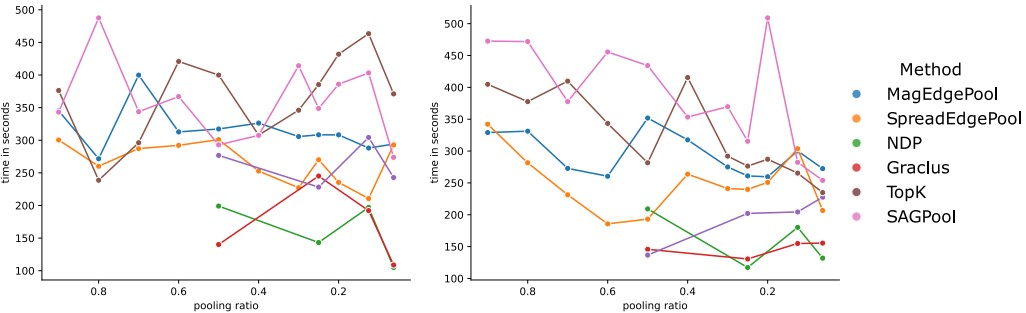

Figure S.3: Runtime comparison for training the GNNs reported in Section 4.1 for NCI1 using different pooling layers. Plots show the mean time in seconds per run using general convolutional layers (left) or GIN layers (right).

## D.3 Node Feature Preservation and Expressivity

Graph pooling should not only preserve graph structure, but also preserve relevant node feature information during pooling. That is, pooling is frequently used after initial rounds of message passing and data representations learnt by previous layers should be respected and effectively encoded by the pooling procedure [32]. One way of investigating node feature retention during pooling, is to evaluate how well a graph can be reconstructed from its pooled version. We follow the experimental setup by Grattarola et al. [32] to investigate. In particular, this experiment uses a model architecture, similar to the model proposed in Section 4.1, where each graph gets pooled to around 50% of nodes after the initial MLP and GNN layer. Then, the pooled graphs are up-scaled again by reversing the node selection step used by each pooling layer. From these unpooled graph representations, a further GNN and post-processing MLP layer are trained and the task is set to output the reconstructed node feature representation. This model is trained on each example graph using Adam to minimize the mean squared error (MSE) between the input and output node features with a learning rate of $0.0005$ and early stopping on the training loss with a patience of 1000 epochs and a tolerance of $10^{-6}$. Each experiment is repeated three times across different random seeds. See Grattarola et al. [32] for further explanations on the model architecture and experimental setup.

For this experiment, we expect SpreadEdgePool to perform comparably well as we specifically designed our pooling algorithm so that features are averaged during pooling. Further restricting the number of times a node can be merged effectively prevents the collapse of entire portions of the graph, which aids reconstruction. SpreadEdgePool pooling thus successfully encodes node information while allowing for a flexible choice of pooling ratio. This is confirmed by the results in Table S.7, which highlight that SpreadEdgePool overall performs well at the features reconstruction task, especially for the sensor graph reaching low reconstruction errors. Alternative methods, in particular node drop approaches such as TopK and SAGPool, show notably worse feature preservation during this experiment indicating the benefits of more expressive pooling operations, such as SpreadEdgePool.

Table S.7: Mean and standard deviation of the reconstruction MSE for reconstructing the original node positions from the pooled graph representations for different example graphs and pooling methods. Our proposed algorithm, SpreradEdgePool, does well at faithfully encoding the feature representations.

| | Ring | Sensor | Barbell | Community | Erdős–Rényi | Torus |
|---|---|---|---|---|---|---|
| SpreadEdge | 5.47e-07 ± 2.63e-07 | 2.78e-05 ± 3.04e-07 | 3.42e-04 ± 1.42e-06 | 3.71e-03 ± 5.80e-05 | 6.49e-07 ± 3.33e-07 | 5.29e-07 ± 1.33e-07 |
| NDP | 3.08e-07 ± 3.57e-07 | 4.07e-05 ± 4.57e-06 | 4.54e-04 ± 2.01e-05 | 2.52e-01 ± 9.50e-06 | 1.46e-06 ± 1.18e-06 | 5.68e-07 ± 1.02e-07 |
| Graclus | 6.87e-04 ± 7.56e-07 | 2.67e-06 ± 2.31e-06 | 1.82e-03 ± 3.22e-07 | 2.42e+00 ± 2.11e-04 | 4.76e-02 ± 3.75e-07 | 7.10e-07 ± 8.60e-08 |
| NMF | 4.78e-07 ± 2.95e-07 | 1.96e-05 ± 1.39e-05 | 5.80e-04 ± 4.53e-07 | 6.06e-01 ± 1.43e-04 | 5.04e-07 ± 3.31e-07 | 2.52e-07 ± 2.93e-07 |
| TopK | 1.21e-01 ± 8.23e-03 | 5.83e-03 ± 2.16e-03 | 1.55e-02 ± 1.10e-02 | 6.03e+00 ± 2.21e+00 | 5.30e-03 ± 7.49e-03 | 1.72e-01 ± 8.51e-03 |
| SAGPool | 1.45e-01 ± 2.52e-02 | 2.01e-03 ± 2.73e-03 | 4.12e-02 ± 4.18e-02 | 4.76e+00 ± 1.57e+00 | 9.60e-05 ± 1.35e-04 | 1.88e-01 ± 4.63e-02 |
| DiffPool | 8.63e-06 ± 4.73e-06 | 3.50e-04 ± 8.32e-05 | 6.50e-04 ± 1.01e-06 | 2.14e-01 ± 2.84e-01 | 3.74e-04 ± 1.46e-04 | 5.17e-05 ± 8.90e-06 |
| MinCut | 2.55e-06 ± 2.68e-06 | 6.56e-06 ± 3.86e-06 | 2.35e-06 ± 1.50e-06 | 1.80e-04 ± 2.01e-04 | 1.44e-06 ± 5.24e-07 | 1.49e-06 ± 9.81e-07 |

Figure S.4: Structure preservation measures for the examples in Figure 1. Pooling is repeated for three different random seeds and the annotations report the means and standard deviations of the structure preservation scores.

Note that the results in Table S.7 capture one specific aspect of feature preservation, namely how well the features of specific example graphs can be reconstructed. This experiment does not assess the generalisation capability of pooling layers. Further, the experimental setup assumes that it is relevant to preserve all node features during pooling, which might not be realistic in practice, where the aim of pooling could be to solely encode task-relevant feature representations. Nevertheless, as discussed above, this extended experiment gives evidence to support that our proposed pooling algorithm, SpreadEdgePool, is capable of outputting expressive feature representations and aggregates node features in a faithful manner, which is likely one of the reasons for its high performance in graph classification tasks.

## D.4 Overview Experiment

Expanding on the qualitative comparison between the example graphs in Figure 1, Figure S.4 shows quantitative structure preservation measures for all example graphs and pooling methods. Specifically, we summarise the spectral loss by Grattarola et al. [32], the spectral distance between the normalised graph Laplacians, and the magnitude difference between the pooled and original graphs as further detailed in Section 4.2.

Figure S.4 demonstrates that SpreadEdgePool and MagEdgePool do not only reach low magnitude differences across these three example graphs, they also show comparatively low spectral distances supporting our findings in Section 4.2. Further, we observe that methods that show worse visual preservation of graph structures, such as NMF, TopK, DiffPool, and MinCut, also reach consistently higher magnitude differences and spectral distances supporting our claim that these methods fail to faithfully preserve graph structures during pooling to varying extents.

## D.5 Graph Classification

Table S.8 further reports extended classification results on additional datasets extending on the results shown in Table 1. We chose not to report on these datasets in the main text because they showed fewer and less notable differences between pooling methods. Note that for the open graph benchmark MolHIV dataset, instead of using stratified cross-validation, we evaluate each model across predefined training, test and validation splits and evaluate their performance across 5 random seeds. Further, we

Table S.8: Classification performance of different pooling layers across multiple datasets. For each dataset, the best performing model is marked in bold and models that do not perform significantly different from the best performing model are coloured green.

| Method | MolHIV (AUROC) | MUTAG | COX2 | BZR | BZR_MD | COX2_MD | DHFR_MD | ER_MD | AIDS |
|---|---|---|---|---|---|---|---|---|---|
| No Pooling | 74.4 ± 0.7 | 81.6 ± 6.3 | 83.8 ± 3.8 | 76.1 ± 9.1 | 73.7 ± 5.3 | 70.2 ± 8.1 | 71.3 ± 1.9 | 74.9 ± 0.8 | 99.0 ± 0.1 |
| MagEdge | 64.6 ± 8.7 | 84.0 ± 7.4 | 86.0 ± 7.3 | **88.1 ± 10.0** | 65.9 ± 2.2 | 76.5 ± 9.2 | **77.9 ± 9.4** | 79.9 ± 7.1 | **99.7 ± 0.1** |
| SpreadEdge | 70.1 ± 2.9 | 85.9 ± 7.4 | 85.1 ± 5.6 | 87.4 ± 9.5 | 69.7 ± 10.0 | **77.1 ± 9.0** | 74.6 ± 11.1 | 83.1 ± 4.0 | **99.7 ± 0.1** |
| NDP | 65.2 ± 7.3 | **91.6 ± 2.4** | **86.1 ± 7.1** | 86.3 ± 9.4 | 72.3 ± 11.3 | 73.9 ± 10.6 | 72.2 ± 11.4 | **84.2 ± 1.7** | 99.6 ± 0.1 |
| Graclus | 70.1 ± 5.3 | 87.4 ± 9.5 | 79.2 ± 12.5 | 80.1 ± 7.1 | 72.0 ± 9.6 | 75.4 ± 9.9 | 71.1 ± 12.3 | 81.7 ± 3.9 | 99.5 ± 0.1 |
| NMF | 73.2 ± 1.3 | 84.0 ± 9.4 | 83.8 ± 8.7 | 80.8 ± 8.5 | 69.0 ± 10.2 | 70.4 ± 6.6 | 70.8 ± 8.6 | 80.9 ± 2.0 | 97.0 ± 1.7 |
| TopK | 72.7 ± 1.8 | 81.9 ± 3.9 | 76.4 ± 7.2 | 75.8 ± 8.0 | 68.4 ± 8.5 | 65.9 ± 7.0 | 69.5 ± 3.3 | 74.0 ± 1.0 | 99.3 ± 0.1 |
| SAGPool | **74.3 ± 3.2** | 82.9 ± 2.3 | 76.8 ± 7.8 | 77.8 ± 5.0 | 66.8 ± 7.0 | 67.0 ± 6.4 | 70.5 ± 2.6 | 75.6 ± 1.2 | 99.0 ± 0.1 |
| DiffPool | 72.1 ± 1.0 | 83.3 ± 2.8 | 76.9 ± 6.5 | 76.8 ± 10.1 | 72.3 ± 3.0 | 69.2 ± 2.1 | 67.9 ± 2.6 | 73.4 ± 1.1 | 98.8 ± 0.2 |
| MinCut | 70.3 ± 3.0 | 80.6 ± 3.7 | 79.1 ± 4.4 | 70.8 ± 8.6 | 69.0 ± 5.0 | 70.0 ± 3.7 | 69.7 ± 1.9 | 73.3 ± 1.6 | 99.2 ± 0.1 |

report AUROC as the performance metric for MolHIV because is the suggested evaluation metric for this very imbalanced dataset. All other results are reported as in Table 1 via the mean and standard deviation of the test accuracy across 10-fold stratified cross-validation. In agreement with our main results, we observe that MagEdgePool and SpreadEdgePool constitute high performing general-purpose pooling methods that reach top performance across these extended datasets. In particular, for smaller biological datasets, such as MUTAG, BZR, or COX2, pooling via magnitude or spread notably improves on the GNN that uses no pooling layer, which indicates the beneficial effects of structure-aware pooling for graph learning.

In order to further clarify the classification performance comparison between methods we visualise the ranking of each model choice via a critical difference diagram. This visualisation uses the Friedman test, a non-parametric test for the performance difference between multiple classifiers with repeat measurements, followed by the Nemenyi post-hoc test that facilitates the comparison across all classifiers [17]. In our case, we use the Python package scikit-posthocs [50] to apply this test to the mean performance scores across datasets. The critical difference plot then links all methods that are found to not be statistically different by a horizontal bar. The results reported in Figure S.5 confirm that our methods are amongst the best-performing group of pooling layers for the datasets reported in Table 1. Similarly, Figure S.6 shows the critical difference diagram and mean ranks across all tasks included in Table 1 or Table S.8. This extended ranking further strengthens our finding that MagEdgePool and SpreadEdgePool reach top performance across classification tasks.

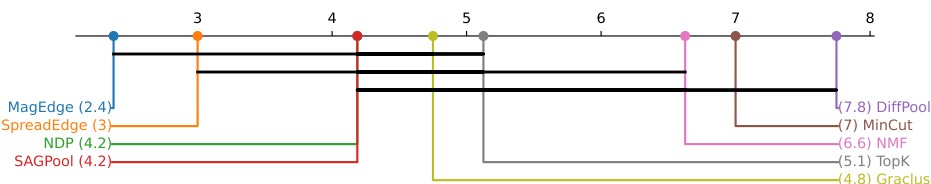

Figure S.5: Critical difference diagram for the classification results from Table 1. Each label corresponds to a choice of pooling method and the method's mean rank across classification tasks. Groups of methods that are found to not be statistically different are linked by horizontal bars.

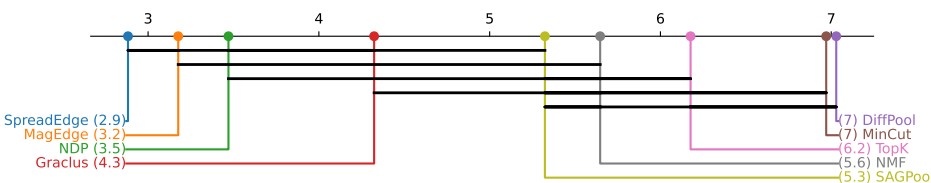

Figure S.6: Critical difference diagram for the classification results from Table 1 and Table S.8. Each label corresponds to a choice of pooling method and the method's mean rank across classification tasks. Groups of methods that are found to not be statistically different are linked by horizontal bars.

Table S.9: Mean and standard deviation of the graph classification accuracy of different pooling methods across datasets.

| Method | ENZYMES | PROTEINS | Mutagenicity | DHFR | IMDB-B | IMDB-M | NCI1 | NCI109 |
|---|---|---|---|---|---|---|---|---|
| No Pooling | 87.3 ± 2.5 | 73.8 ± 0.8 | 80.1 ± 1.3 | 71.4 ± 1.9 | 69.7 ± 0.7 | 46.0 ± 0.7 | 76.5 ± 1.8 | 74.3 ± 2.0 |
| MagEdge | 91.5 ± 3.2 | 76.4 ± 3.9 | 77.5 ± 2.7 | 88.0 ± 3.8 | 72.4 ± 1.7 | 47.4 ± 1.7 | 72.7 ± 2.4 | 73.0 ± 3.3 |
| SpreadEdge | 92.8 ± 1.6 | 75.1 ± 3.1 | 76.0 ± 4.0 | 90.7 ± 3.8 | 71.8 ± 1.5 | 47.3 ± 1.7 | 73.4 ± 2.5 | 71.8 ± 1.8 |
| EdgePool | 92.1 ± 1.4 | 75.8 ± 4.6 | 75.3 ± 3.1 | 83.2 ± 1.0 | 70.9 ± 4.5 | 47.6 ± 3.3 | 71.9 ± 3.0 | 72.7 ± 2.6 |
| Random | 88.1 ± 2.2 | 73.7 ± 1.1 | 73.1 ± 2.1 | 82.1 ± 3.3 | 70.1 ± 0.7 | 45.6 ± 0.8 | 71.1 ± 2.2 | 68.0 ± 3.0 |

## D.6 Comparison with EdgePool

To extend our evaluation, we compare with the `pytorch_geometric` implementation of Edge-Pool [20, 21]. To do so, we implement the GNN architecture specified in Section 4.1 in `PyTorch` and repeat our main experiment reported in Table 1. We find that there is no significant difference in performance between our methods and EdgePool when computed on e.g. IMDB-M, where EdgePool reaches an accuracy of 47.6 ± 3.3, or on IMDB-B, where EdgePool reaches an accuracy of 70.9 ± 4.5. As further reported in Table S.9, similar patterns hold across datasets with our proposed pooling methods reaching comparable accuracies to EdgePool across datasets. As a major advantage, our methods have notably lower computational costs during training than EdgePool as reported in Appendix D.2 and permit choosing flexible pooling ratios. Our methods thus make edge-contraction pooling scalable to larger datasets and enable significantly faster GNN training. On IMDB-B for example, EdgePool takes 362.2 seconds and 666.6 MB for 200 epochs during training, but our method SpreadEdgePool only requires 62.7 seconds and 283.4 MB. We thus find that learning feature-based edge scores as done by EdgePool, is not necessary to ensure classification performance, but rather adds a notable computational burden.

## D.7 Comparison with Randomised Pooling

Research by Mesquita et al. [44] has demonstrated that randomised ablations of Graclus and DiffPool using random cluster assignment can reach competitive performance to established pooling methods on frequently used graph classification datasets, such as IMDB-B, PROTEINS, NCI109, DD and MOLHIV. Questioning standard assumptions of hierarchical pooling methods, Mesquita et al. [44] thus question the utility of localised graph pooling. We find that this discussion and the observations by Mesquita et al. [44] are closely connected to the following challenges:

- Graph learning tasks do not necessarily need the graph structure to reach high performance [7, 15].
- Evaluating the performance of pooling layers is dependent on the specific GNN architecture and benchmarking practices used [24, 44].
- Randomised modifications of expressive pooling operators can retain the expressivity of the underlying pooling operation [8].

Hence, if all task-relevant information is already captured by the learnt features before pooling, randomised baselines can reach competitive performance. Nevertheless, destroying the structure of graphs during pooling can reduce the effectiveness of preceding message-passing layers [8]. Our pooling approach aims to mitigate this risk by presenting geometry-aware and expressive edge-pooling methods.

As an ablation study, we add randomised edge pooling to our main experiment (cf. Table 1) by randomly merging pairs of nodes into super-nodes to pre-compute the pooling assignment and averaging their features during training. We find that for NCI109 randomised pooling reaches an accuracy of 68.0 ± 3.0 as compared to 71.8 ± 1.8 for SpreadEdgePool. On NCI1 random pooling reaches an accuracy of 71.1 ± 2.2 compared to 73.4 ± 2.5 for SpreadEdgePool. Thus, the random baseline performs worse than any alternative pooling method on NCI109 and decreases the accuracy on NCI1 slightly. We hypothesise that this decrease is due to graph structure being relevant for these learning tasks [15] and our pooling method better preserves structural properties. In contrast, we observe fewer performance differences for other standard datasets, such as ENZYMES, PROTEINS, IMDB-B, or IMDB-M. This confirms the findings by Coupette et al. [15] that on these datasets GNNs can reach competitive accuracies even when using completely randomised adjacencies as inputs. We thus find that an expressive aggregation of the node features during pooling can suffice [8] and preserving graph structure might not be necessary for solving these tasks to begin with.

## D.8 Preserving Graph Structure

Extending on the results reported in Figure 3 and Section 4.2, we further report the distribution of structure preservation measures across pooling ratios. Specifically, Figure S.8 shows line plots that summarise the mean magnitude difference relative to the original graph and the normalised spectral distance between all original and pooled graphs from the NCI1 dataset in the leftmost column. The remaining plots illustrate the quantiles of the same measures split up per pooling method. Overall, these individual plots support our assessment that MagEdgePool and SpreadEdgePool consistently reach low magnitude differences and comparably low spectral distances, with these trends being more pronounced in terms of the relative difference in magnitude after pooling. Further, we repeat the experiment reported in Figure 3 for further datasets, specifically for DHFR, PROTEINS, and ENZYMES and summarise the results in Figure S.7. We observe consistent trends across these examples that support the results reported for NCI1 in Figure 3. Specifically we find that our proposed pooling methods, MagEdgePool and SPreadEdgePool, reach the lowest magnitude differences across pooling ratios and datasets, and that this corresponds to low spectral distances between the pooled and original graphs.

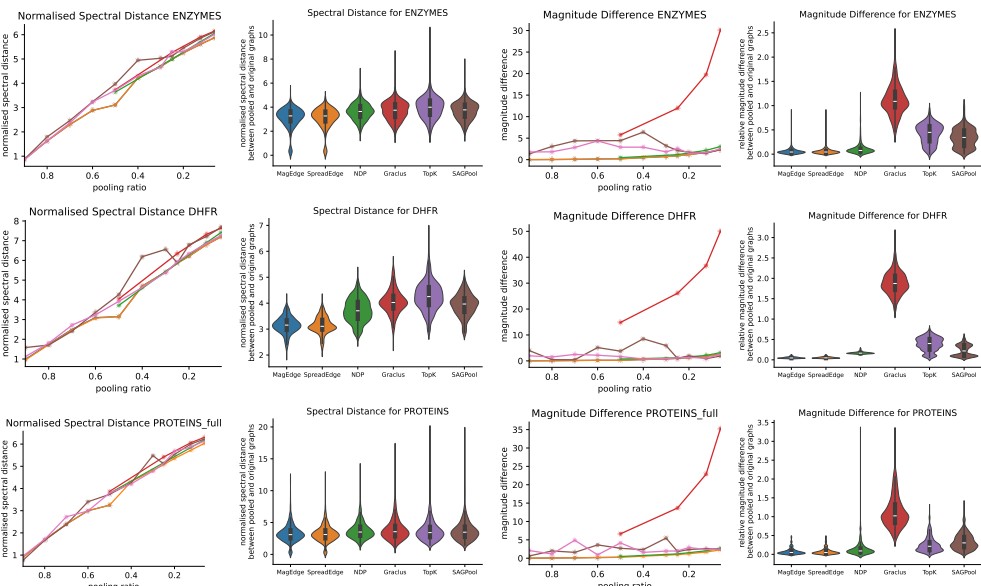

Figure S.7: Structure preservation for all graphs in the ENZYMES, DHFR, and PROTEINS datasets across pooling ratios. Left: The spectral distance between the normalised Laplacians of the original and the pooled graphs. Right: The relative difference in magnitude, summarising proportional differences in structural diversity after pooling. Violin plots show the variability across graphs at pooling ratio 0.5.

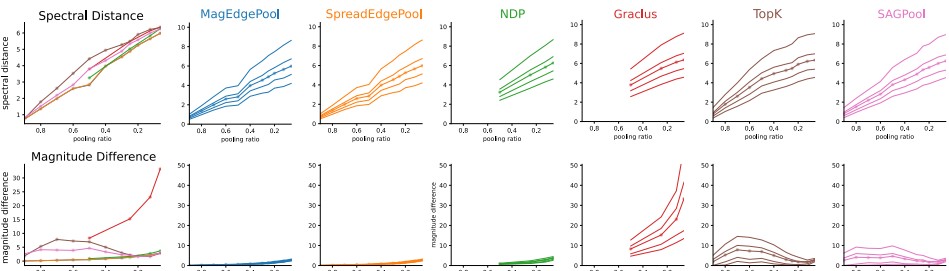

Figure S.8: Structure preservation for the NCI1 dataset across pooling ratios. Top row: The spectral distance between the original and pooled graphs. Bottom row: The relative difference in magnitude. Bold lines show the mean values of each score across graphs and thin lines the 10%, 25%, 75% and 90% quantiles.

