# OpenReview forum: "Geometry-Aware Edge Pooling for Graph Neural Networks"
_NeurIPS.cc/2025/Conference — NeurIPS 2025 poster_

### Official Review · Reviewer_GecT · 2025-07-01

**Clarity:** 3
**Significance:** 3
**Originality:** 2
**Rating:** 3
**Confidence:** 4

**Summary:**

The study proposes a novel magnitude and spread-guided edge pooling layer, incorporating metric space diversity measures into graph compression for the first time. This approach effectively preserves graph geometric structures while enhancing classification performance. The paper provides rigorous theoretical foundations, including error propagation bounds for both magnitude and spread metrics.

**Questions:**

1. **Runtime and Memory Usage:** Could the authors provide detailed runtime and memory usage data? Comparisons with other training-free methods (e.g., TopK and MinCut) would be valuable.

2. **Scalability to Large-Scale Graphs:** Current experiments focus on small graphs. Testing on large-scale datasets, such as PDB for Enzyme Commission (EC) number prediction or Gene Ontology (GO) term prediction [1], would better demonstrate practical applicability.

3. **Extension to Regression Tasks:** The study currently focuses exclusively on classification. Could the proposed method also be extended to regression tasks, such as Protein Localization Prediction on UniProt [2]?

[1] José Juan Almagro Armenteros et al., DeepLoc: Prediction of protein subcellular localization using deep learning. Bioinformatics, 33 (21):3387–3395, 2017. https://www.nature.com/articles/s41467-021-23303-9#Sec27

[2] Vladimir Gligorijević et al., Structure-based protein function prediction using graph convolutional networks. Nature Communications, 12(1):3168, 2021. https://pubmed.ncbi.nlm.nih.gov/29036616/

**Ethical Concerns:**

["NO or VERY MINOR ethics concerns only"]

**Final Justification:**

I have revised my score upward after noticing that my initial assessment was based on a misreading of the performance table. Nevertheless, I remain concerned about the very high scalability of the proposed method, especially in the context of large-scale graph data.

**Limitations:**

Experiments are confined to small graphs and classification tasks, lacking validation on large-scale datasets or regression problems.
While computational complexity is mentioned, concrete runtime measurements are absent.

**Paper Formatting Concerns:**

No obvious formatting errors

**Quality:**

2

**Strengths And Weaknesses:**

Strengths:

1. **Improved Classification Accuracy:** Achieves superior performance on most datasets, with significant improvements noted on specific datasets like DHFR.

2. **Robustness:** Demonstrates minimal performance fluctuations across a wide range of pooling ratios (6.25% to 100%), indicating strong adaptability for dynamic graph compression needs.

3. **Interpretability:** Decision-making based on geometric invariants offers transparent and explainable results.

Weaknesses:

1. **Inconsistent Performance:** Although beneficial for most datasets, the proposed method underperforms on certain datasets (e.g., Mutagenicity), with only half achieving state-of-the-art (SOTA) results.

2. **High Computational Complexity:** Despite being training-free, PCA computations introduce considerable complexity, potentially limiting scalability to large-scale graphs.

3. **Lack of Novelty in Diffusion Distance:** The diffusion distance concept applied by the authors is not novel; rather, the contribution lies in its specific application to graph pooling.

4. **Spelling Errors:** The manuscript contains several spelling errors (e.g., "MagEgdePool" instead of "MagEdgePool" on line 195; repeated "SpreadEgdePool" errors on lines 220, 229, and 341), suggesting the need for careful proofreading.

---

> ### Author Rebuttal · Authors · 2025-07-30
>
> **Summary**
>
> **New experiments with preliminary results and additions:**
> - *Computational efficiency:* Evaluate runtime and memory usage across pooling methods. Report and contrast pre-training and training costs. Highlight the computational improvements compared to EdgePool.
> - *Regression experiments:* Report and compare the performance of our pooling method at regression tasks during revisions.
>
> **Clarifications and revisions:**
> - *SOTA results:* Clarify the purpose of graph pooling. Contextualise the reported accuracy gains w.r.t. to baseline pooling methods and existing literature [1, 2].
> - *Novelty of diffusion distances:* Explain the conceptual benefits of using diffusion distances on graphs. Highlight potential extensions of our methods based on established theory on diffusion maps [4, 5].
> - *Typos:* Correct minor typos in the spelling of MagEdgePool.
>
> **Extended discussion:**
> - *Computational efficiency:* Add an extended discussion on the computational efficiency of our pooling method in comparison to alternative methods.
> - *Large-graphs tasks:* Discuss potential extensions to large-scale graphs.
>
> ---
>
>
> We would like to clarify the points raised by Reviewer GecT.
>
> > Improved Classification Accuracy: Achieves superior performance on most datasets, with significant improvements noted on specific datasets like DHFR.
>
> We are thankful that this review acknowledges the superior classification accuracy and robust performance achieved by our pooling methods.
>
> >    Inconsistent Performance: Although beneficial for most datasets, the proposed method underperforms on certain datasets (e.g., Mutagenicity), with only half achieving state-of-the-art (SOTA) results.
>
> **The purpose of our evaluation is not to achieve SOTA results across all datasets and all potential GNN architectures**, but to give a consistent benchmark that enables a **fair comparison of pooling methods across tasks**. For that purpose we plug each pooling layer into the same GNN architecture as commonly done to compare pooling methods. However, it is unrealistic to believe that this setup could or should achieve SOTA results on every datasets without dataset-specific modifications.
>
> Nevertheless, our results show consistent and robust performance across tasks, and reach very high accuracies above what is reported in most pooling benchmarks [1, 2]. In fact, **on Mutagenicity our pooling method outperforms all alternative pooling layers** included in our comparison. For this reason, we argue that **our pooling methods reach SOTA results amongst baseline pooling methods** and do not underperform.
>
> >    High Computational Complexity: Despite being training-free, PCA computations introduce considerable complexity, potentially limiting scalability to large-scale graphs.
>
> We would like to clarify that **our pooling method does not compute PCA**. We compute the magnitude or spread of a graph as defined in Equations (1) and (2) for edge pooling and discuss the computational complexity of these calculations in our manuscript. However, this is not equivalent to PCA. This seems to be a misunderstanding about the computation of our algorithm.
>
> Further, we provide additional results on the memory cost and computational complexity in our response to AV7X. We thus find that **the computational performance of our proposed pooling method is competitive** for most standard graph classification datasets. Overall, our pooling approach notably improves on the computational efficiency of EdgePool. Thus, we **address scalability limitations of existing (edge) pooling methods**, such as EdgePool, DiffPool, and MinCutPool, which scale notably worse to large-scale graphs.
>
>
> >    Runtime and Memory Usage: Could the authors provide detailed runtime and memory usage data? Comparisons with other training-free methods (e.g., TopK and MinCut) would be valuable.
>
> We would like to clarify that **TopK and MinCut are trainable rather than training-free methods**. Considering the distinction between, trainable (TopK, MinCut, SAGPool, DiffPool) and training-free methods (ours, Graclus, NDP, NMF), **we are happy to provide runtime and memory comparisons** in the response to reviewer AV7X.
>
> Specifically, we find that e.g. on IMDB-M our method SpreadEdgePool (99.7s per run) allows for notably faster training than MinCutPool (132.8s) and moderately faster training than TopKPool (110.9s). Thus, we find that **the improved computational efficiency of GNN training is an advantage of our pooling method**.
>
>
> >    Lack of Novelty in Diffusion Distance: The diffusion distance concept applied by the authors is not novel; rather, the contribution lies in its specific application to graph pooling.
>
>
> Our pooling method indeed leverages the diffusion distance, a well-studied metric on general measure spaces that leverages both geometrical and topological aspects of the input data. The **novelty of our method** lies in its formulation as a technique based on metric space magnitude, a recently introduced invariant for metric spaces. As reviewer 7pUv pointed out, we are the **first paper to study magnitude-informed approaches in the context of graph learning**. The **use of the diffusion distance**, which users can replace if they need to, **is rather a strength of our method**.
>
> For further research, we believe that the usage of diffusion distances offers the chance to leverage a rich theory on approximation methods via landmarks [4], or localised diffusion computations [5], which can lead to further computational improvements and extension of our methods. We thus believe the established theory on diffusion maps is a benefit of our methodology.
>
>
> >    Spelling Errors: The manuscript contains several spelling errors (e.g., "MagEgdePool" instead of "MagEdgePool" on line 195; repeated "SpreadEgdePool" errors on lines 220, 229, and 341), suggesting the need for careful proofreading.
>
> We thank the reviewer for pointing out specific typos and look forward to correcting them in the final manuscript. We are happy to rectify these minor mistakes in our revision.
>
>
> >    Scalability to Large-Scale Graphs: Current experiments focus on small graphs. Testing on large-scale datasets, such as PDB for Enzyme Commission (EC) number prediction or Gene Ontology (GO) term prediction [1], would better demonstrate practical applicability.
>
> Experiments on large-scale graphs would indeed be of interest, but truly massive graphs are rarely encountered for graph classification tasks [3] and offer a particular set of challenges regarding memory and computational costs. **Existing pooling methods** that process dense adjacency matrices, such as MinCut or DiffPool, **are unlikely to scale to such tasks** without major modifications [2, 3].
>
> Given these necessary architecture adjustments, **large-scale experiments remain outside the scope of our work for now.** However, for future work, we note that our edge pooling method could indeed be scaled to larger graphs by e.g. computing the edge scores in a localised or mini-batched manner on carefully chosen subgraphs rather than on the whole graph. The sparse pre-trained edge assignment could then be used efficiently for training on large-scale graphs.
>
> We also note that already, **our pooling method notably outperforms EdgePool or dense pooling methods such as MinCut or DiffPool in terms of memory costs and computational efficiency**, improving the scalability of edge pooling in most practical applications to standard graph classification tasks.
>
>
>
> >    Extension to Regression Tasks: The study currently focuses exclusively on classification. Could the proposed method also be extended to regression tasks, such as Protein Localization Prediction on UniProt [2]?
>
> We would like to note that **protein localisation prediction on UniProt is a node classification task** rather than a regression task. Working with the suggested datasets, PDB or UniProt, would require domain-specific preprocessing and careful evaluation beyond what is feasible in the scope of rebuttals.
>
> **Our pooling architectures can be extended to either node classification or graph regression problems.** However, we consider graph-level tasks to be the most relevant applications of pooling methods [1] and thus choose to focus on graph classification tasks for our analysis.
>
> Overall, we hope that our answer could clarify any misunderstandings.
>
>
> **References**
>
> [1] Wang, P., et al., 2024. A comprehensive graph pooling benchmark: Effectiveness, robustness and generalizability.
>
> [2] Grattarola, D., et al., 2022. Understanding pooling in graph neural networks.
>
> [3] Bianchi, F.M. and Lachi, V., 2023. The expressive power of pooling in graph neural networks.
>
> [4] Long, A.W. and Ferguson, A.L., 2019. Landmark diffusion maps (L-dMaps): Accelerated manifold learning out-of-sample extension.
>
> [5] David, G. and Averbuch, A., 2012. Hierarchical data organization, clustering and denoising via localized diffusion folders.

---

> > ### Comment · Reviewer_GecT · 2025-08-04
> >
> > > Inconsistent Performance
> > Although approximately half of the experiments did not achieve the top results, the overall ranking demonstrates a reasonable performance advantage. Considering this, I have decided to increase my score.
> >
> > > Scalability to Large-Scale Graphs
> > Even with the proposed approximations (indicated by rows marked with *), MagEdgePool and SpreadEdgePool continue to require significantly more computational time and memory compared to traditional methods when applied to larger datasets. The primary bottleneck is the eigen-decomposition step necessary for computing diffusion distances.
> >
> > > Novelty
> > Utilizing magnitude/spread as a global diversity metric to guide edge contractions is indeed novel and clearly distinguishes this work from existing methods that solely rely on attention-based scores. Nevertheless, since this approach still fundamentally depends on a similarity matrix, it retains conceptual similarities to prior similarity-based pooling techniques like SimPool. The authors should highlight these differences and similarities more clearly in the paper.

---

> > > ### Author Response · Authors · 2025-08-04
> > >
> > > We thank the reviewer for their careful consideration and for increasing their score. We agree with the conclusion that the overall performance shows reasonable advantages, and with the scalability limitations regarding the distance computations. We thank the reviewer for highlighting related similarity-based pooling methods, and will discuss the differences and similarities to such methods in more detail in revisions.

---

> > > ### Comment · Area_Chair_2bj9 · 2025-08-05
> > >
> > > Reviewer GecT,
> > >
> > > Could you please elaborate on your thoughts on the authors’ rebuttal? It seems that some reviewers have become more positive about the paper after reading it. In your view, is scalability really a concern here? For example,  it could be okay for a new method to work only on moderately sized graph if the performance is good, and scaling to trillions of nodes although nice, can come a little later? Or do you think the approach is not at all practical as it stands?
> > >
> > > I’d also appreciate clarification on your comment that “since this approach still fundamentally depends on a similarity matrix, it retains conceptual similarities to prior similarity-based pooling.” In particular, could you be more precise on why using similar tools or data structures implies a lack of innovation in this context? Aren't we all just building off the same data and tools? Why is this here too similar?

---

### Official Review · Reviewer_rNzm · 2025-07-02

**Clarity:** 3
**Significance:** 3
**Originality:** 3
**Rating:** 5
**Confidence:** 4

**Summary:**

This paper introduces MagEdgePool and SpreadEdgePool, two novel geometry-aware edge pooling methods for GNNs that address the challenge of reducing graph size while preserving structural information. Unlike existing pooling approaches that focus on node clustering or dropping, these methods use edge contraction guided by the magnitude and spread of metric spaces - measures that quantify a graph's structural diversity based on diffusion distances. The key insight is that edges whose removal minimally affects the graph's magnitude or spread are the most redundant and should be collapsed first, thereby preserving the graph's geometric properties during pooling. The authors provide theoretical analysis showing that their methods are isometry-invariant and preserve key properties like additivity for disjoint graphs.

**Questions:**

1. In the abstract, I am confused about this sentence: "However, existing pooling operations often optimise for the learning task at the expense of fundamental graph structures and interpretability."
2. Is the proposed method in this paper effective for the graph regression task?

**Ethical Concerns:**

["NO or VERY MINOR ethics concerns only"]

**Final Justification:**

I have no further questions. I vote for acceptance.

**Limitations:**

yes

**Paper Formatting Concerns:**

No concern

**Quality:**

3

**Strengths And Weaknesses:**

Strengths:
1. The paper provides rigorous mathematical grounding with theorems on additivity, isomorphism invariance, and bounds relating magnitude and spread, giving the methods solid theoretical justification beyond empirical results.
2. The proposed methods achieve top rankings across 8 diverse datasets, demonstrating robustness and generalizability rather than being tuned to specific graph types.

Weaknesses:
1. In some parts, the graph is defined as G = (X, E), while in some other parts it is G = (V, E). Besides, the definitions of X, V, and E are not given.
2. In the visualization results from Figure 1, the pooling ratio of different pooling methods are not consistent.

---

> ### Author Rebuttal · Authors · 2025-07-30
>
> **Summary**
>
> **New experiments with preliminary results and additions:**
> - *Regression experiments:* Report and compare the performance of our pooling method at regression tasks during revisions.
>
> **Clarifications and revisions:**
> - *Definition of a graph:* Correct the definition of a graph to be consistent across the manuscript.
> - *Changes in the abstract:* Clarify the sentence in the abstract.
> - *Figure 1:* Double-check the number of nodes. Specify that graphs are pooled to approximately half their size.
>
> ---
>
>
> We thank the reviewer for their positive feedback, in particular for highlighting that our pooling approach demonstrates robust and generalisable performance.
>
> > In some parts, the graph is defined as G = (X, E), while in some other parts it is G = (V, E). Besides, the definitions of X, V, and E are not given.
>
> > In the abstract, I am confused about this sentence: "However, existing pooling operations often optimise for the learning task at the expense of fundamental graph structures and interpretability."
>
> We thank the reviewer for pointing out specific revisions in the text and **we will clarify the definition of a graph and reformulate the sentence in the abstract**. Specifically, the sentence refers to the observation that existing trainable pooling operators tend to optimise for a task-specific loss. However, such methods might not preserve the structure of the graph during pooling and make counterintuitive choices on which graph properties to preserve. This is illustrated by the comparison between pooling methods in Figure 1, which shows that pooling methods such as TopKPool, SAGPool, DiffPool, or MinCutPool do not necessarily lead to interpretable graph coarsening.
>
>
> > In the visualization results from Figure 1, the pooling ratio of different pooling methods are not consistent.
>
> The pooling ratios in Figure 1 are consistently set to 50%. However, some pooling methods (e.g. Graculus or NDP) do not give exact control over the number of vertices, but pool graphs to **approximately half** of their original size. This is why the exact number of nodes differ slightly. We will clarify this in the manuscript and aim to provide a revised version of Figure 1 with more comparable node numbers whenever possible.
>
> >  Is the proposed method in this paper effective for the graph regression task?
>
> Our method can be extended to regression tasks and flexibly plugged into any GNN. For revisions, **we plan to incorporate regression tasks into our analysis** and adjust our model architecture and evaluation to fit these tasks. **We expect regression performance to agree with the classification results.** The arguments why our proposed edge pooling methods perform well at graph classification, namely that features are aggregated in an expressive manner and that graph structure is preserved during pooling, similarly hold for graph regression tasks.

---

> > ### Comment · Reviewer_rNzm · 2025-08-05
> >
> > Thank you for your detailed rebuttal and clarifications. I have no further questions.

---

### Official Review · Reviewer_AV7X · 2025-07-02

**Clarity:** 4
**Significance:** 3
**Originality:** 3
**Rating:** 5
**Confidence:** 3

**Summary:**

The paper "Geometry-Aware Edge Pooling" focuses on leveraging diffusion geometry to develop pooling layers that preserve both the metric structure and structural diversity of graphs in graph neural networks. The authors introduce two novel pooling methods, MagEdgePool and SpreadEdgePool, which outperform existing pooling layers across various graph classification tasks while retaining key spectral properties and maintaining high accuracy across different pooling ratios.

**Questions:**

- Is the code open-source and reproducible?
- Can you provide more experimental results regarding the efficiency of the method?

**Ethical Concerns:**

["NO or VERY MINOR ethics concerns only"]

**Final Justification:**

The authors should also address the overlooked scalability issue and enhance the analysis by comparing with global pooling methods such as SOPool and DKEPool to make the study more complete.

**Limitations:**

Yes

**Quality:**

3

**Strengths And Weaknesses:**

Strengths

- The paper proposes novel pooling methods that consider graph geometry and structural diversity, leading to improved performance compared to existing approaches.

- The empirical results demonstrate the superiority of MagEdgePool and SpreadEdgePool over alternative pooling layers across diverse graph classification tasks.

- The paper is well-organized and easy to follow. Relevant theoretical proofs have been conducted.



Weaknesses：

- The experimental results demonstrating the efficiency of the method should be presented.
- The method becomes more challenging to apply as the size of the graph nodes increases.

---

> ### Author Rebuttal · Authors · 2025-07-30
>
> **Summary**
>
> **Extended experiments with preliminary results and additions:**
> - *Computational efficiency:* Evaluate runtime and memory usage across pooling methods. Report and contrast pre-training and training costs.
> - *Computational speed-ups:* Propose and benchmark approximation techniques for estimating diffusion distances to speed up graph pooling.
>
>
> **Extended discussion:**
> - *Computational efficiency:* Discuss further strategies for ensuring computational scalability.
> - *Reproducibility and usability:* Include details on the reproducibility and open-source implementation of our pooling method.
>
> ---
>
>
> We thank the reviewer for their positive feedback and concrete questions, which we are happy to address. Specifically, we are thankful for the review highlighting the rigorous mathematical and theoretical grounding and strong performance of our proposed pooling method.
>
>
> > Can you provide more experimental results regarding the efficiency of the method?
>
> We will include **extended runtime and memory comparisons** in our revisions and want to present experimental results on the computational efficiency of our pooling method. Specifically, we train the GNN architecture specified in our manuscript using GIN layers across 200 epochs using 10-fold stratified cross-validation. We compare our edge pooling methods (MagEdgePool , SpreadEdgePool ) to trainable pooling methods from torch_geometric (EdgePool, TopKPool, SAGPool) or from torch-geometric-pool (DiffPool, MinCutPool). We record the mean and standard deviation of the runtimes in seconds and GPU memory usage in MB per cross-validation run.
>
> We note that in particular our proposed edge pooling methods **MagEdgePool and SpreadEdgePool allow for significantly more efficient GNN training than EdgePool**. Similarly, our methods generally improve on the runtimes for dense pooling methods such as DiffPool and MinCutPool, and improve on the computational efficiency of TopKPool and SAGPool.
>
> Table of **training times in seconds compared across pooling methods**. The fastest methods are marked in bold.
>
> | Method         |      DHFR       |     ENZYMES     |     NCI109      |  Mutagenicity   |  IMDB-BINARY   |  IMDB-MULTI    |
> |----------------|------------------|------------------|------------------|------------------|----------------|----------------|
> | MagEdgePool    | 39.8 ± 6.0       | **22.0 ± 0.2**   | 186.0 ± 9.1      | **180.3 ± 29.0** | 75.4 ± 1.4     | **89.6 ± 6.1**  |
> | SpreadEdgePool | **34.8 ± 1.0**   | 23.0 ± 0.7       | **181.3 ± 29.4** | 206.7 ± 20.6     | **62.7 ± 0.9** | 99.7 ± 4.3     |
> | EdgePool       | 184.1 ± 4.0      | 209.3 ± 4.1      | 807.4 ± 22.2     | 790.7 ± 22.1     | 362.3 ± 3.4    | 392.7 ± 11.9   |
> | TopKPool       | 50.7 ± 0.8       | 24.1 ± 0.6       | 226.1 ± 21.3     | 243.8 ± 15.4     | 88.2 ± 1.1     | 110.9 ± 2.2    |
> | SAGPool        | 54.4 ± 1.3       | 25.7 ± 0.4       | 250.4 ± 25.7     | 253.0 ± 20.0     | 92.6 ± 2.1     | 131.1 ± 1.3    |
> | DiffPool       | 52.2 ± 6.8       | 36.4 ± 3.3       | 242.4 ± 12.5     | 274.0 ± 21.5     | 106.7 ± 5.5    | **74.9 ± 2.0** |
> | MinCutPool     | **33.8 ± 2.5**   | 28.2 ± 1.7  | 210.3 ± 34.4     | 201.0 ± 3.7      | 93.2 ± 1.6     | 132.8 ± 3.2    |
>
>
> Table of **memory usage in MB compared across pooling methods**. The most efficient methods are marked in bold.
> | Method         |      DHFR       |     ENZYMES     |     NCI109      |  Mutagenicity   |  IMDB-BINARY   |  IMDB-MULTI    |
> |----------------|------------------|------------------|------------------|------------------|----------------|----------------|
> | MagEdgePool    | **84.0 ± 0.1**   | **91.2 ± 2.5**   | **97.0 ± 1.1**   | **94.8 ± 1.0**   | **283.4 ± 28.0** | **197.6 ± 17.8** |
> | SpreadEdgePool | **84.0 ± 0.1**   | **90.6 ± 2.3**   | **96.8 ± 1.0**   | **95.2 ± 1.0**   | **283.4 ± 28.0** | **197.4 ± 17.8** |
> | EdgePool       | 125.6 ± 10.4     | 148.4 ± 15.0     | 118.6 ± 7.7      | 118.0 ± 7.4      | 666.6 ± 106.6   | 526.2 ± 89.2    |
> | TopKPool       | 89.6 ± 9.0       | 94.4 ± 4.0       | 105.0 ± 1.4      | 105.8 ± 6.5      | **259.4 ± 31.5** | **193.8 ± 17.5** |
> | SAGPool        | 104.2 ± 0.6      | 94.2 ± 3.8       | 105.2 ± 1.4      | 107.8 ± 8.1      | **273.4 ± 30.1** | **209.0 ± 17.9** |
> | DiffPool       | 90.2 ± 10.2      | 94.6 ± 2.5       | 103.8 ± 1.1      | 299.8 ± 31.9     | **258.0 ± 40.3** | 240.8 ± 26.4    |
> | MinCutPool     | 90.6 ± 10.3      | 94.2 ± 2.4       | 103.6 ± 0.8      | 288.4 ± 38.3     | **258.0 ± 40.3** | **196.6 ± 26.0** |
>
>
> Having observed that pre-computed edge-pooling speeds up GNN training times, we further compare with the **runtime and memory costs of other non-trainable pooling operators prior to training**. We report the computational costs of computing the pooling assignment for all graphs in the datasets.
>
> As reported in our manuscript, we find that **SpreadEdgePool is generally more efficient than MagEdgePool**. Beyond the exact computations of our pooling methods, we also approximate the change in diffusion distance during edge contraction by updating the contracted node’s distances to be the minimum distance to the original nodes. These approximate versions denoted MagEdgePool\* and SpreadEdgePool\* notably reduce computational costs.
>
> Table of **pre-training times in seconds compared across non-trainable pooling methods**. The fastest method is marked in bold. The fastest approximation of our pooling method is marked in cursive.
>
> | Method           |   DHFR   | ENZYMES | Mutagenicity | NCI109  | IMDB-BINARY | IMDB-MULTI |
> |------------------|----------|---------|---------------|---------|--------------|-------------|
> | MagEdgePool      |   61.9   |  197.4  |    4765.1     |  678.8  |    801.8     |    480.2    |
> | MagEdgePool*     |   24.6   |   21.0  |     81.3      |   74.9  |     77.4     |     69.9    |
> | SpreadEdgePool   |   66.7   |   68.3  |    316.1      |  208.8  |    287.2     |    243.8    |
> | SpreadEdgePool*  | *10.9*   | *16.0*  |   *52.3*      | *55.6*  |   *56.0*     |   *66.0*    |
> | NDP              |   5.6    |   4.1   | **37.1**      |  32.6   |     5.8      |     7.0     |
> | NMF              |   7.8    |  10.8   |    39.0       |  32.1   |     8.7      |     9.0     |
> | Graclus          | **3.8**  | **2.9** |    54.5       | **18.2**|  **4.5**     |  **6.0**    |
>
> Table of **pre-training memory usage in MB compared across non-trainable pooling methods**. The most efficient method is marked in bold. The fastest approximation of our pooling method is marked in cursive.
>
> | Method           |   DHFR   | ENZYMES | Mutagenicity | NCI109  | IMDB-BINARY | IMDB-MULTI |
> |------------------|----------|---------|---------------|---------|--------------|-------------|
> | MagEdgePool      |   76.2   |  163.8  |    158.9      |  179.0  |    162.1     |    93.1     |
> | MagEdgePool*     |   83.7   | *34.8*  |   *47.0*      | *72.2*  |   *11.5*     |    65.2     |
> | SpreadEdgePool   |   61.9   |  60.1   |    177.9      |  93.1   |    204.1     |    75.8     |
> | SpreadEdgePool*  | *33.6*   |  54.1   |    60.5       |  72.9   |    53.2      |  *23.8*     |
> | NDP              | **3.0**  | **5.0** |   **7.0**     |  13.4   |     5.1      |     6.5     |
> | NMF              |   3.8    |   5.9   |    39.9       | **10.5**|     4.2      |     5.9     |
> | Graclus          |  15.8    |   7.5   |    17.5       |  59.5   |  **1.0**     |  **1.5**    |
>
>
>
> The presented results show one of the limitations discussed in our manuscript, namely that our edge pooling approach scales in the number of edges as well as the size of the graphs. Nevertheless, our proposed pooling methods still notably **outperform EdgePool in terms of computational efficiency** when considering both the preprocessing and training costs as reported above. Further, the pretraining costs of our method range in a number of seconds, only need to be computed once per dataset, and the memory requirements remain below what is required by GNN training. Hence, we find that **our proposed pooling methods scale sufficiently well to standard graph datasets**.
>
> We look forward to presenting extended experiments on computational efficiency in revisions. For future work, we believe there is a strong potential to further speed up our pooling methods by investigating alternative graph metrics, testing sampling heuristics to restrict the edge score computations to a subset of candidate edges, or computing edge score on local subgraphs rather than on the whole graph.
>
>
> >  Is the code open-source and reproducible?
>
> We will publish a light-weight implementation of our pooling method as an open-source package to allow users to try our pooling method on their own datasets.
>
> This is in addition to our supplementary code submission, which can be used to reproduce our experiments.

---

> ### Comment · Reviewer_AV7X · 2025-08-02
> **Rebuttal Response**
>
> Thank you for the response. However, it appears that the author has overlooked the second weakness regarding scalability with the increasing number of graph nodes.  Moreover, it is valuable to include a comparison and discussion of global pooling methods such as SOPool [1] and DKEPool [2], etc., as this would make the analysis more comprehensive.
>
> [1] Wang, Z., & Ji, S. (2023). Second-order pooling for graph neural networks. *IEEE Transactions on Pattern Analysis and Machine Intelligence*, 45(6), 6870–6880.
>  [2] Chen, K., Song, J., Liu, S., et al. (2023). Distribution knowledge embedding for graph pooling. *IEEE Transactions on Knowledge and Data Engineering*, 35(8), 7898–7908.

---

> > ### Author Response · Authors · 2025-08-02
> > **Rebuttal Response**
> >
> > We thank the reviewer for their comment and clarification. Scalability regarding the number of nodes is discussed in our manuscript when detailing the theoretical complexity and the limitations of our pooling methods. For revisions, we will include extended experiments on the computational costs of processing graphs of increasing sizes in comparison with alternative pooling methods. Hence, we will discuss and give practical recommendations on the maximum size of the graphs that can practically be processed with our algorithm. Further, we will add a discussion on global pooling methods to our manuscript, such as the ones suggested by the reviewer, and compare with the goal and purpose of hierarchical pooling methods. We are happy to answer any additional questions the reviewer might have.

---

### Official Review · Reviewer_7pUv · 2025-07-03

**Clarity:** 3
**Significance:** 3
**Originality:** 3
**Rating:** 4
**Confidence:** 3

**Summary:**

This paper proposes a novel structure-aware edge-pooling method that reduces the size of input graphs by collapsing redundant edges. The authors view a graph as a metric space (with a clever choice of metric) and use the mathematical concept of magnitude to measure the importance of each edge for the graph’s geometry. Furthermore, they introduce the notion of the spread of a metric space as an alternative to magnitude, offering better computational properties while also being applicable to the proposed pooling method. Both approaches demonstrate strong performance on graph classification tasks.

**Questions:**

1. The DHFR dataset was the single case where an expressive improvement was obtained. Do the authors have an explanation for this?

2. As pointed out by the authors, Figure 3 indicates that preserving magnitude corresponds to lower spectral distances and better retention of spectral properties. Would you say this also corresponds to the lower difference between the MagEdgePool/SpreadEdgePool and No Pooling results reported in Table 1? Since Figure 3 only shows the results for the NCI1, could report the same analyses for another dataset such as DHFR?

3. In general, do you have an intuition/experiment/theorem that explicitly relates the magnitude/spread (or magnitude’s difference between input and reduced graph) with the accuracy?

**Ethical Concerns:**

["NO or VERY MINOR ethics concerns only"]

**Final Justification:**

After the discussion, my remaining concerns are:

(i) the claims of superior performance in the manuscript are overstated (c.f. comparison with EdgePool);

(ii) there is a chance that this method is just as "good" as random pooling based on prior criticism of pooling methods --- and there is not enough evidence in this work to support otherwise.

I feel (i) can be easily solved by softening the claims in this work. Furthermore, even with (ii) on the way, I appreciate the methodological novelty the method and still lean towards acceptance.

**Limitations:**

Please check "Weaknesses" and "Questions" above.

**Paper Formatting Concerns:**

No particular formatting concerns

**Quality:**

3

**Strengths And Weaknesses:**

## Strengths

1. The use of magnitude by the machine learning community is very recent and the authors are the first to introduce it in the context of graph learning. As far as I know, it is also the first time that the mathematical notion of spread is used in machine learning, contributing to previous (and future) works that use magnitude since it is a faster alternative.

2. Presents a novel and structure-preserving form to reduce graphs.

## Weaknesses:

1. The authors say that EdgePool [1] is the most successful edge-based. Although they argue its major drawback of pooling graphs to half their size, there is no experimental comparison to support the superior performance of MagEdgePool/SpreadEdgePool with another edge-pooling method.

2. The author cites [2] but they do not debate their argument that pooling methods (e.g GRACLUS and DIFFPOOL) are not significantly superior to baselines built upon randomized cluster assignments. A more thorough discussion and inclusion of randomized baselines in the experiments would improve the paper.

3. The accuracy reported on Table 1 presents only slightly better results, i.e, the experimental results are not convincing.

[1] F. Diehl. Edge contraction pooling for graph neural networks

[2] Diego Mesquita, Amauri H. Souza, Samuel Kaski, Rethinking pooling in graph neural networks

---

> ### Author Rebuttal · Authors · 2025-07-30
>
> **Summary**
>
> **Extended experiments:**
> - *Comparison with EdgePool:* Extend experiments with EdgePool on IMDB-M, IMDB-B, and further datasets. Highlight runtime and memory efficiency improvements.
> - *Randomised baselines:* Add randomised edge pooling experiments (already partly reported for NCI1, NCI109). Expand to further datasets.
> - *Spectral preservation:* Reproduce Figure 3 analyses for further datasets to confirm consistent magnitude/spectral distance patterns.
>
> **Clarifications and revisions:**
> - *Positioning vs. EdgePool:* Emphasise that the main advantages are computational scalability and flexible pooling ratios, not raw accuracy gains.
> - *Accuracy interpretation:* Clarify that pooling aims at efficiency + structure preservation; contextualise improvements with stratified 10-fold CV and prior baselines.
> - *Link between magnitude/spread and accuracy:* Elaborate theoretical reasoning behind structure preservation; connect to expressivity results in existing work [3].
>
> **Extended discussion:**
> - Add a dedicated subsection on *expressivity vs. structure preservation*.
> - Incorporate recent findings [2,3,4,6] to contextualise when structural preservation matters.
> - Discuss conditions under which randomised baselines can remain competitive, clarifying limits of structural information use.
>
> ---
>
> We thank the reviewer for the insightful and detailed questions and find they offer a great opportunity to discuss the reasons behind the robust performance of our pooling methods.
>
> > The authors say that EdgePool [1] is the most successful edge-based. Although they argue its major drawback of pooling graphs to half their size, there is no experimental comparison to support the superior performance of MagEdgePool/SpreadEdgePool with another edge-pooling method.
>
> To extend our evaluation, we compare with the pytorch_geometric implementation of EdgePool. We find that **there is no significant difference in performance between our methods and EdgePool** when computed on e.g. IMDB-M, where EdgePool reaches an accuracy of 47.6  ± 3.3; and 70.9  ± 4.5 on IMDB-B. We expect this same pattern to hold across further datasets and will report extended results during revisions.
>
> As a major advantage, **our methods have notably lower computational costs during training than EdgePool** (cf. the response to reviewer AV7X). Our methods thus make edge-contraction pooling scalable to larger datasets and enable significantly faster GNN training. On IMDB-B for example, EdgePool takes 362.2 seconds and 666.6 MB for 200 epochs during training, but our method SpreadEdgePool only requires 62.7 seconds and 283.4 MB.
>
> We thus find that learning feature-based edge scores as done by EdgePool, is not necessary to ensure classification performance, but rather adds a notable computational burden. The main advantage of our pooling methods over EdgePool is thus improved scalability as well as the flexible choice of pooling ratio.
>
> > The author cites [2] but they do not debate their argument that pooling methods (e.g GRACLUS and DIFFPOOL) are not significantly superior to baselines built upon randomized cluster assignments.
>
> We thank the reviewer for their insightful suggestion and **we will include further discussion and randomised ablations to the paper**.
>
> We find that this discussion and the observations by [2] are closely connected to the following challenges:
> Standard graph learning **tasks do not necessarily require the graph structure to ensure high performance** [4, 6].
> Randomised modifications of expressive pooling operators can retain the **expressivity of the underlying pooling operation** [3].
>
> Hence, if all task-relevant information is already captured by the learnt features before pooling, randomised baselines can reach competitive performance. Nevertheless, destroying the structure of graphs during pooling can hinder the effectiveness of the preceding message passing layers [3]. Our pooling approach aims to mitigate exactly this risk by presenting **geometry-aware and expressive edge-pooling methods**.
>
> As preliminary results, we add randomised edge pooling to our main experiment reported in Table 1. We find that for NCI109 randomised edge pooling reaches an accuracy of 68.0 ± 3.0 as compared to 71.8 ± 1.8 for SpreadEdgePool. On NCI1 random edge pooling reaches an accuracy of 71.1 ± 2.2 compared to 73.4 ± 2.5 for SpreadEdgePool. Thus, the **random baseline performs worse than any alternative pooling method on NCI109** and decreases the accuracy on NCI1 slightly. We hypothesise that this decrease in accuracy is observed because graph structure is relevant for these learning tasks [4] and our pooling method better preserves structural properties.
>
> > The accuracy reported on Table 1 presents only slightly better results, i.e, the experimental results are not convincing.
>
> The accuracies in Table 1 are computed for GNNs whose only difference is the choice of pooling layer to allow for a **comprehensive and fair comparison across pooling methods**. In this context, it is in fact a **challenging task to achieve very high performance gains solely based on the choice of pooling strategy**. We therefore argue that the goal of pooling is not necessary to beat any non-pooling baseline, but to enable more efficient training while preserving task-relevant structure or feature information.
>
> Given we use stratified 10-fold cross validation to ensure a fair comparison, the reported accuracies are comparable to the baseline performance reported by [5] and **similar or higher than the degree of improvement reported by established pooling methods**, such as EdgePool [1]. Thus, our results show robust performance and realistic accuracy gains.
>
> We are happy to extend our experimental setup with additional comparison partners or datasets in case the reviewer has specific suggestions.
>
> > The DHFR dataset was the single case where an expressive improvement was obtained. Do the authors have an explanation for this?
>
> DHFR most benefits from the regularising effect of structure-aware edge pooling due to two potential factors. First of all, DHFR shows structural and geometric differences between graphs representing active and inactive compounds. Our pooling method helps preserve key graph structures, such as rings and graph connectivity, during pooling as visualised in Figure 2 in our manuscript. This can aid the preceding layers’ capability to discriminate between the coarsened graphs. Second, our pooling method gives an expressive pooling operator as discussed above. Amongst other expressive operators, we hypothesise that the aggregation via edge collapse pooling might be most suitable for this task specifically.
>
> > Figure 3 indicates that preserving magnitude corresponds to lower spectral distances and better retention of spectral properties. Would you say this also corresponds to the lower difference between the MagEdgePool/SpreadEdgePool and No Pooling results reported in Table 1? Since Figure 3 only shows the results for the NCI1, could report the same analyses for another dataset such as DHFR?
>
> **Yes, we can reproduce the results from Figure 3 for other graph datasets**, such as DHFR, and will report further datasets as ablations in the revisions. The same patterns we report for NCI1 hold for other datasets in the sense that we can consistently show that MagEdgePool and SpreadEdgePool induce both low spectral distances and low differences in magnitude during pooling indicating superior preservation of the graph structure.
>
> We argue that the more **faithful preservation of graph properties does contribute to the better performance** reported in Table 1. This is particularly relevant for classification tasks such as NCI1 that rely on the geometry of the input graph for task performance [3, 4].
>
> > In general, do you have an intuition/experiment/theorem that explicitly relates the magnitude/spread (or magnitude’s difference between input and reduced graph) with the accuracy?
>
> There is a strong **link between preserving magnitude/spread during pooling and preserving the graph structure** of the input graph. Magnitude can be expressed as a polynomial with variables given by the eigenvalues of the similarity matrix Z = [exp(-d(xi,xj))], which we compute from the diffusion distances d. We believe that under mild conditions small changes in the Laplacian matrix lead to small changes in the similarity matrix which then induce small spectral distances between the input and the pooled graph. We aim to analyse this relationship further in future work.
>
> Further, we find **theoretical links between the expressivity of pooling operators and the accuracies** reported in Table 1. Edge contraction pooling when for example using sum pooling for aggregating the features fulfills sufficient conditions laid out by [3] for preserving the expressive power of the preceding message passing layers. Thus our pooling methods fulfil theoretical guarantees for both expressive feature retention as well as graph structure preservation.
>
> We hypothesise that structure preservation will be particularly beneficial in case labels (that is, tasks) are directly related to the structure. As recent research shows [6], this prerequisite is not necessarily given, warranting careful study. We will add an extended discussion on expressivity and structure-preservation during revisions.
>
> **References**
>
> [1] Diehl, F., et al., 2019. Towards graph pooling by edge contraction.
>
> [2] Mesquita, D., et al., 2020. Rethinking pooling in graph neural networks.
>
> [3] Bianchi, F.M. and Lachi, V., 2023. The expressive power of pooling in graph neural networks.
>
> [4] Coupette, C., et al., 2025. No Metric to Rule Them All: Toward Principled Evaluations of Graph-Learning Datasets.
>
> [5] Grattarola, D., et al., 2022. Understanding pooling in graph neural networks.
>
> [6] Bechler-Speicher, M., et al., 2024. Graph neural networks use graphs when they shouldn't.

---

> > ### Comment · Reviewer_7pUv · 2025-08-05
> >
> > Dear authors, thank you for your clarifications and additional experiments.
> >
> > After reading the rebuttal, It seems to me that your claim in the abstract at "achieve superior performance compared to alternative pooling layers across a range of diverse graph classification tasks" is somehow overstate --- c.f. comparison against EdgePool.
> >
> > I would also like to see more evidence to really infer your method is better than random pooling. I find experiments only on NCI1 and NCI109 to be too little to conclude so.

---

> > > ### Author Response · Authors · 2025-08-06
> > >
> > > We thank the reviewer for their comments!
> > >
> > > Concerning the wording in the abstract, we will adjust this to make it less ambiguous. Our primary point was that magnitude- and spread-based pooling performs very well on average (as can be seen by the mean rank column in the table). When comparing with `EdgePool`, we reported preliminary experiments, which (so far) appear to show _similar_ pooling performance (for the specific setting), albeit at a much higher computational cost.
> > >
> > > ## Proposed updates
> > >
> > > - We will run more such experiments and add a better context for the comparison. In particular, we will compare performance over pooling ratios wherever possible (similar to Figure 3).
> > >
> > > - We will add _critical difference plots_ (see Demšar, *Statistical Comparisons of Classifiers over Multiple Data Sets*, Journal of Machine Learning Research 7 (2006) 1–30), enabling us to discuss _statistically significant_ performance differences. A preliminary calculation already shows that our proposed methods are part of the best-performing group of methods. We will use the additional datasets to strengthen this claim.
> > >
> > > - We will indeed, as outlined above, add randomised baselines for all datasets.
> > >
> > > We thank the reviewer very much for their support and look forward to discussing any additional concerns!

---

### Note · Authors · 2025-08-12

As a final remark, we wanted to raise that we enjoyed this rebuttal period. We found it to be very productive and insightful; amidst all the noise and confusion about the reviewing process on social media, we felt it appropriate to express our gratitude. We are confident that the suggestions made by the reviewers will serve to further improve our work and its impact.

---

### Decision · Program_Chairs · 2025-09-17

**Decision:**

Accept (poster)

**Comment:**

This paper introduces two graph pooling layers for Graph Neural Networks (GNNs), MagEdgePool and SpreadEdgePool, designed to preserve fundamental graph structures during pooling by using edge collapses. These methods leverage diffusion geometry and are guided by magnitude and spread, which are measures of structural diversity, to iteratively reduce a graph's size while maintaining its metric structure. Empirical results show that these approaches outperform existing pooling layers on various graph classification tasks, effectively preserve key spectral properties of the input graphs, and maintain high accuracy across different pooling ratios.


Overall, the reviewers all agreed that the method is elegant, novel and seems to work well. One reviewer (GecT) recommends rejection (3), but upon examination of their review, the reviewer (a) has made a lot of errors in their review (e.g. referring to PCA whilst PCA is not mentioned) and (b) has not appropriately engaged in the discussion. Consequently, we will down weight this reviewer's contribution. Most reservations from the other reviewers concerned the computational efficacy of the method, which was addressed (to a satisfactory extent) by the author during the rebuttal phase.  The authors have also provided new experiments during the rebuttal. The reviewers also expressed their interest in seeing other relevant benchmarks that were not included in the manuscript. Should this paper be accepted, these would constitute interesting additions to the camera ready version of the paper.